# Relative effects of open biomass and crop straw burning on haze formation over central and eastern China: modelling study driven by constrained emissions

Khalid Mehmood[1*], Yujie Wu[1*], Liqiang Wang[1], Shaocai Yu[1,2+], Pengfei Li[1,3+], Xue Chen[1], Zhen Li[1], Yibo Zhang[1], Mengying Li[1], Weiping Liu[1], Yuesi Wang[4], Zirui Liu[4], Yannian Zhu[5], Daniel Rosenfeld[6], and John H. Seinfeld[2]

[1]Research Center for Air Pollution and Health; Key Laboratory of Environmental Remediation and Ecological Health, Ministry of Education, College of Environment and Resource Sciences, Zhejiang University, Hangzhou, Zhejiang 310058, P.R. China
[2]Division of Chemistry and Chemical Engineering, California Institute of Technology, Pasadena, CA 91125, USA.
[3]College of Science and Technology, Hebei Agricultural University, Baoding, Hebei 071000, P.R. China
[4]State Key Laboratory of Atmospheric Boundary Layer Physics and Atmospheric Chemistry, Institute of Atmospheric Physics, Chinese Academy of Sciences, Beijing 100029, China
[5]Meteorological Institute of Shaanxi Province, 36 Beiguanzhengjie, Xi'an 710015, China
[6]Institute of Earth Sciences, The Hebrew University of Jerusalem, Jerusalem, Israel

[*]These authors contributed equally to this work.

[+]*Correspondence to*: Shaocai Yu (shaocaiyu@zju.edu.cn); Pengfei Li (lpf_zju@163.com)

**To be submitted to**

**Atmospheric Chemistry and Physics**

**Abstract.** Open biomass burning (OBB) has large potential in triggering local and regional severe haze with elevated fine particulate matter ($PM_{2.5}$) concentrations and could thus deteriorate ambient air quality and threaten human health. Open crop straw burning (OCSB), as a critical part of OBB, emits abundant gaseous and particulate pollutants, especially in fields with intensive agriculture, such as central and eastern China (CEC). This region includes nine provinces, i.e., Hubei, Anhui, Hunan,

Jiangxi, Shandong, Jiangsu, Shanghai, and Fujian. The former four ones are located inland, while the others are on the eastern coasts. However, uncertainties in current OCSB and other types of OBB emissions in chemical transport models (CTMs) lead to inaccuracies in evaluating their impacts on haze formations. Satellite retrievals provide an alternative that can be used to simultaneously quantify emissions of OCSB and other types of OBB, such as the Fire INventory from NCAR version 1.5 (FINNv1.5), which, nevertheless, generally underestimate their magnitudes due to unresolved small fires. In this study, we

selected June in 2014 as our study period, which exhibited a complete evolution process of OBB (from June 1 to 19) over CEC. During this period, OBB was dominated by OCSB in terms of the number of fire hotspot and associated emissions (74 ~ 94 %), most of which were located at Henan and Anhui (> 60 %) with intensive enhancements from June 5 to 14 (> 80 %). OCSB generally exhibits spatiotemporal correlation with regional haze over the central part of CEC (Henan, Anhui, Hubei, and Hunan), while other types of OBB emissions had influences on Jiangxi, Zhejiang, and Fujian. Based on these analyses, we

establish a constraining method that integrates ground-level $PM_{2.5}$ measurements with a state-of-art fully coupled regional meteorological and chemical transport model (the two-way coupled WRF-CMAQ) in order to derive optimal OBB emissions based on FINNv1.5. It is demonstrated that these emissions allow the model to reproduce meteorological and chemical fields over CEC during the study period, whereas the original FINNv1.5 underestimated OBB emissions by 2 ~ 7 times, depending on specific spatiotemporal scales. The results show that OBB had substantial impacts on surface $PM_{2.5}$ concentrations over

CEC. Most of the OBB contributions were dominated by OCSB, especially in Henan, Anhui, Hubei, and Hunan, while other types of OBB emissions also exerted influence in Jiangxi, Zhejiang, and Fujian. With the concentration-weighted trajectory (CWT) method, potential OCSB sources leading to severe haze in Henan, Anhui, Hubei, and Hunan were pinpointed. The results show that the OCSB emissions in Henan and Anhui can cause haze not only locally but also regionally through regional transport. Combining with meteorological analyses, we can find that surface weather patterns played a cardinal role in

reshaping spatial and temporal characteristics of $PM_{2.5}$ concentrations. Stationary high-pressure systems over CEC enhanced local $PM_{2.5}$ concentrations in Henan and Anhui. Then, with the evolution of meteorological patterns, Hubei and Hunan in the low-pressure system were impacted by areas (i.e., Henan and Anhui) enveloped in the high-pressure system. These results suggest that policymakers should strictly undertake interprovincial joint enforcement actions to prohibit irregular OBB, especially OCSB over CEC. Constrained OBB emissions can, to a large extent, supplement estimations derived from satellite

retrievals as well as reduce overestimates of bottom-up methods.

## 1 Introduction

Open biomass burning (OBB) has adverse impacts on ambient air quality and human health, owing to the fact that OBB generally emits abundant gaseous and particulate pollutants in a short period of time, particularly carbonaceous aerosols (e.g., black carbon (BC) and organic carbon (OC)) (Li et al., 2010; Rose et al., 2010; Cheng et al., 2013; Ding et al., 2013; Cheng et al., 2014; Saleh et al., 2014; Washenfelder et al., 2015; Vakkari et al., 2018; Bikkina et al., 2019). It has been estimated that OBB contributes approximately 65 % of global annual average primary OC emissions (Bond et al., 2013), and more than 40 % of fine particulate matter ($PM_{2.5}$) concentrations in specific cases of regional haze (Zhang and Cao, 2015; Long et al., 2016; Gao et al., 2016; Sun et al., 2016; Li et al., 2017). In China, estimated growth rates of BC, OC and primary $PM_{2.5}$ emitted by OBB from 2002 to 2016 were 180 %, 191 %, and 192 %, respectively (Mehmood et al., 2018), suggesting an urgent need to control OBB emissions.

Open crop straw burning (OCSB), as a crucial part of OBB, generally occurs on a large spatial scale during the harvest seasons in regions with intensive agriculture activities, because it is still the most effective, efficient and economical measure to dispose of open crop straw (Li et al., 2007;Qin and Xie, 2011; Zhang et al., 2017; Zhuang et al., 2018; Xu et al., 2019; Zhang et al., 2019). Previous studies showed that emissions from OCSB accounted for more than 80% of those from OBB over China during the past decade. More importantly, emissions increased steadily during the past decade, thereby directly causing or substantially exacerbating regional haze (Wu et al., 2018).

China is experiencing frequent and severe regional haze, partly resulting from substantial and extensive OBB (Bi et al., 2010; Huang et al., 2014; Andersson et al., 2015;Zhang et al., 2015; Zhang et al., 2015a; Hong et al., 2016; Chen et al., 2017; An et al., 2019). Open crop straw burning in China accounted for around 20 % of global production and rose with an average annual rate of 4 % during the past decade (Bi et al., 2010; Hong et al., 2016). Specifically, central and eastern China (CEC), as a cardinal granary in the world, was associated with large quantities of crop planting and thus played a crucial role in intensive OCSB, which has been suspected of causing rapid increases of $PM_{2.5}$ concentrations in this region during the harvest seasons (Yamaji et al., 2010; Wang et al., 2015; Zhang et al., 2016; Ding et al., 2016; Liu et al., 2018; Wang et al., 2018; Yu et al., 2019). Therefore, it is necessary to understand the effects of OBB and OCSB on haze formation over CEC in order to provide effective regulations on mitigating OBB activities.

Current studies using chemical transport models (CTMs) indicate, however, that high uncertainties exist in accurately quantifying impacts of OBB on a regional scale. First is the challenge of representing the magnitude and spatiotemporal distribution of OBB and OCSB emissions, despite newly developed OBB emission inventories during the past decade (Streets et al., 2003;Liu et al., 2015; Zhang and Cao, 2015; Zhang et al., 2015a; Li et al., 2016; Qiu et al., 2016; Zhou et al., 2017; Zhao et al., 2017; Liu et al., 2018a; Liu et al., 2018b; Zhang et al., 2019; Dai et al., 2019). Traditional bottom-up statistical methods cannot capture rapid outbreaks of OBB and OCSB and fail to produce reliable emission amounts and distributions. Moreover, few studies have applied CTMs to evaluate these bottom-up emissions. Also, satellite-based top-down emission inventories cannot generally resolve small fire hotspots (usually around large ones) and can underestimate OBB and OCSB

emissions by 2 ~ 20 times (Wiedinmyer et al., 2011; Randerson et al., 2012; Uranishi et al., 2019). Several studies have attempted to improve the performance of CTMs by increasing OBB or OCSB emissions with a uniform proportion regardless of specific spatial and temporal scales (Chuang et al., 2015; Pimonsree et al., 2018; Uranishi et al., 2019). Recent studies used observed and simulated constraints to optimize OBB emissions (Hooghiemstra et al., 2012; Konovalov et al., 2014; Yang and Zhao, 2019). Nevertheless, this method still did not represent the spatiotemporal heterogeneity of uncertainties in OBB emissions and has yet to be applied in CEC. Therefore, combining sufficient observed and simulated results, as well as satellite-based fire information, could provide the prospective alternative to optimize OBB emissions. To support related policy-making more effectively, potential major OBB emission sources and corresponding meteorological drivers should also need to be identified.

Instantaneous emissions from OBB and OCSB are also poorly co-constrained because of the difficulty of distinguishing them with high spatial and temporal resolution (Yan et al., 2006; Tsao et al., 2012; Lei et al., 2013; Zhang et al., 2013; Cheng et al., 2013; Monks et al., 2015; Laing et al., 2016; Chen et al., 2017; Wu et al., 2018; Hamilton et al., 2018; Lee et al., 2018; Li et al., 2019; Uranishi et al., 2019; Yang and Zhao, 2019). Consequently, most of the previous studies focused only on individual biomass (e.g., corn, wheat, and rice) or total OBB. Hence, few findings could isolate spatial and temporal influences of OCSB from those of OBB in China, especially over CEC (Li et al., 2007; Cao et al., 2008; Fu et al., 2012; Cheng et al., 2014; Huang et al., 2014; Long et al., 2016; Zhang et al., 2019). Satellite observations by NASA's moderate resolution imaging spectroradiometer (MODIS) or visible infrared imaging radiometer (VIIRS) offer an attractive alternative by providing fire information (e.g., burned areas, fire locations) that is able to distinguish OCSB from total OBB. Further, these observations can serve as an computational basis to yield products for their respective emissions, such as the Fire INventory from NCAR version 1.5 (FINNv1.5) (Wiedinmyer et al., 2011).

Comprehensive quantitative analyses of relative effects of OBB and OCSB on haze formation is of critical significance for CEC. In this study, we focus on June in 2014, when abundant $PM_{2.5}$ with exceptionally high concentrations (e.g., the hourly peak surface $PM_{2.5}$ concentration > 200 μg m$^{-3}$) enveloped the CEC, potentially due to effects of intense OBB and OCSB emissions during this harvest season. To understand the evolution of regional haze over CEC, we analyzed spatial and temporal characteristics in the OBB and OCSB emissions in FINNv1.5, ground-measured $PM_{2.5}$ concentrations, and satellite-based aerosol optical depths (AODs). We have established a constraining method that integrates abundant ground-level $PM_{2.5}$ measurements with a state-of-art fully coupled regional meteorological and chemical transport model (i.e., the two-way coupled WRF-CMAQ) in order to derive optimal OBB emissions based on the original FINNv1.5 emissions. With these constrained emissions, the model reproduced spatiotemporal variations in chemical and meteorological fields. Further, we applied the model to simultaneously quantify the relative contributions of OCSB and OBB emissions to surface $PM_{2.5}$ concentrations. The backward trajectory (HYSPLIT) and concentration-weighted trajectory (CWT) methods were integrated to pinpoint potential major OCSB sources. Corresponding meteorological patterns resulting in this regional haze event were also analysed. Finally, we compare the constrained OBB emissions as well as their associated effects with previous studies.

## 2 Methods and data

### 2.1 The two-way coupled WRF-CMAQ model

We utilize the two-way coupled Weather Research and Forecasting (WRF) model and Community Multiscale Air Quality

(CMAQ) model (i.e., the two-way coupled WRF-CMAQ model, hereinafter as the WRF-CMAQ model) (Wong et al., 2012; Yu et al., 2013) to simulate meteorological and chemical fields over CEC (Yu et al., 2018). This fully-coupled model represents a significant advancement over the offline WRF-CMAQ system (Byun and Schere, 2006), since the former expands to encompass aerosol-radiation interactions to which OBB and OCSB should be closely related (Wang et al., 2014; Huang et al., 2016; Baró et al., 2017; Li et al., 2017b; Singh et al., 2018; Malavelle et al., 2019). Moreover, the newly developed coupler

helps to improve the consistency between WRF and CMAQ with regard to meteorological characteristics (Wong et al., 2012). The WRF-CMAQ model was configured with the CB05 and AERO6 schemes for gas and aerosol chemistry simulations, respectively (Yarwood et al., 2005; Carlton et al., 2010); thus primary emissions (e.g., primary OC, BC, and dust) and secondary pollutants (e.g., secondary sulfate, nitrate, ammonium, and organic aerosols) are both considered. Aerosols are described by three modes (Aitken, Accumulation, and Coarse) with a lognormal distribution (Seinfeld and Pandis, 2016).

Additionally, the ISORROPIA II model was applied to perform thermodynamic equilibrium calculations for $K^+$–$Ca^{2+}$–$Mg^{2+}$– $NH_4^+$–$Na^+$–$SO_4^{2-}$–$NO_3^-$–$Cl^-$–$H_2O$ aerosol systems among gas, liquid and particulate phases (Fountoukis and Nenes, 2007). In terms of meteorology simulations, we selected the two-moment Morrison cloud microphysics scheme (Morrison and Gettelman, 2008), the Kain-Fritsch cumulus cloud parameterization scheme (KF2) (Kain, 2004), the Rapid Radiative Transfer Model for General circulation models (RRTMG) (Clough et al., 2005), the Pleim-Xiu land surface scheme (PX) (Xiu and

Pleim, 2001) and the asymmetric convective model (ACM2) (Pleim, 2007a; Pleim, 2007b) for the cloud physics, radiative transfer, land surface energy balance and planetary boundary layer simulations, respectively.

Meteorological initial and lateral boundary conditions were derived from the ERA interim reanalysis dataset operated by the European Centre for Medium-Range Weather Forecasts (ECMWF) with spatial resolution of 1 ° × 1 °and temporal resolution of 6 hours (http://www.ecmwf.int/products/data, last access: 2 August 2019). The CMAQ-default initial and boundary

chemical conditions were used. A spin-up period of seven days was used to minimize the influence of initial chemical conditions (Liu et al., 2010; Wang et al., 2012; Liu et al., 2018b). To eliminate numerical artifacts that commonly occur in WRF external boundary relaxation zones, we trimmed off seven grid cells on each edge of the domain (Wong et al., 2012; Yu et al., 2012). In addition, meteorological fields were reinitialized by reanalysis data every 48 hours in order to constrain corresponding simulated results (Lo et al., 2008; Zhao et al., 2010; Zhang et al., 2016b).

In this study, the Multi-resolution Emission Inventory for China version 1.2 (MEICv1.2) (http://www.meicmodel.org, last access: 2 August 2019) mainly implemented by Tsinghua University was used for anthropogenic emissions (Li et al., 2015a). This inventory comprises five anthropogenic sectors (i.e., industry, power plants, residential and transport and agriculture) for $PM_{2.5}$ and its major precursors (e.g., CO, $SO_2$, $NO_X$, primary $PM_{2.5}$). It also builds a framework to speciate non-methane volatile organic compounds (NMVOCs) and in accordance with the CB05 mechanism. Anthropogenic emissions outside China

were derived from the Task Force Hemispheric Transport of Air Pollution version 2 (HTAPv2) (Janssens-Maenhout et al., 2015; Li et al., 2017b), which includes two additional sectors (i.e., aircrafts and ships). These two additional sectors were then aggregated to the transport sector so as to remain compatible with MEICv1.2 with regard to anthropogenic sectors. Unlike off-line anthropogenic emissions, natural sources for biogenic and dust emissions were calculated inline using the Biogenic Emission Inventory System version 3.14 (BEISv3.14) (Carlton and Baker, 2011) and a windblown dust scheme embedded in CMAQ (Choi and Fernando, 2008), respectively.

To examine OBB and OCSB impacts over CEC, we conducted the simulations in a domain covering most of China with a 12 km horizontal resolution (Fig. 1). There were 31 sigma-pressure vertical layers ranging from the surface to the top pressure of 100 hPa, 20 of which are located below around 3 km to achieve finer meteorological and chemical characterization within the planetary boundary layer. It should be noted that CEC in this study comprises 10 provinces, including Anhui (AH), Hubei (HB), Henan (HEN), Hunan (HUN), Shandong (SD), Zhejiang (ZJ), Jiangsu (JS), Shanghai (SH), Fujian (FJ) and Jiangxi (JX), which are underlined with black thick outlines in Fig. 1.

To comprehensively validate the model performance, we would evaluate the spatial distributions of model-derived AODs, besides primary chemical and meteorological factors. Theoretically, not only particles but also gases have the ability to attenuate the intensity of light. AODs, generally severing as the feature of extinctions, should be the combined function of their scattering and absorption. However, owing to the insignificant magnitude of gases, we focused only on particles to estimate the model-derived AODs as the following equations (Malm et al., 1994; Binkowski and Roselle, 2003; Song et al., 2008; Park et al., 2011; Jeon et al., 2016):

$$AOD_{MODEL} = \sum_{i=1}^{N} (\sigma_{sp} + \sigma_{ap}) \Delta Z_i, \tag{1}$$

$$\sigma_{sp} = 0.003f(RH)(NH_4^+ + +SO_4^{2-} + NO_3^-) + 0.004OM + 0.001FS + 0.0006CM, \tag{2}$$

$$\sigma_{ap} = 0.01LAC, \tag{3}$$

where i denoted to the vertical layer number and $Z_i$ referred to the corresponding layer thickness. The OM, FS, CM, and LAC were the mass concentrations of organic species, fine soil, coarse particles, and black carbon, respectively and uniformly configured with the units of mg/m$^3$. Their respective scattering and absorbing coefficients (i.e., 0.003, 0.004, 0.001, 0.0006, and 0.001) were recorded in m$^2$/mg. The f(RH) represented the aerosol growth factor that was estimated based on the relative humidity. All relevant parameters were extracted from the model results.

**2.2 OBB and OCSB emissions**

In this study, we used FINNv1.5 to characterize spatiotemporal features of OCSB and other types of OBB emissions. FINNv1.5 provides an unique chance to characterize spatial and temporal estimations of trace gas and particle emissions from seven types of OBB, including savannas, grasslands, woody savannas, shrublands, tropical forests, temperate forests, boreal forests, and croplands (Wiedinmyer et al., 2011). Its distinctive advantages include global coverage, high temporal and spatial resolutions (daily and 1km), and adequate land use types, mainly due to the utilization of MODIS NRT active fire products (MCD14DL,

https://firms.modaps.eosdis.nasa.gov, last access: 5 August 2019), which are processed with the standard MOD14/MYD14 fire and thermal anomalies. Nevertheless, several critical weaknesses originating from satellite retrievals are present, leading to OBB emissions that are largely underestimated (Chuang et al., 2015; Pimonsree et al., 2018; Uranishi et al., 2019). Some previous studies indicated that the actual total amount of primary $PM_{2.5}$ emissions over Northeast China was nearly 20 times higher than that estimated by FINNv1.5 (Uranishi et al., 2019). Additionally, the lack of local emission information (e.g., local biomass loading data, emission factors) could introduce extra uncertainties. Therefore, FINNv1.5 utilized in this study should require further adjustments to achieve more accurate estimates for OBB and OCSB emissions.

It is important to represent injection heights of OBB in CTMs, which could significantly affect its regional transport. Previous studies demonstrated that different injection heights could lead to distinct $PM_{2.5}$ responses (Freitas et al., 2006; Leung et al., 2007) and pressure-weighted injections within the troposphere is a reliable alternative (Hyer et al., 2007). In this study, we determined heights of the hourly top ($P_{top}$) and bottom ($P_{bottom}$) of the OBB plume using a quick plume rise model (Eq. (4) and Eq. (5)). This process was calculated based on the buoyant efficiency ($B$) available from the corresponding hourly and size class tables (Tables S1 and S2) (Tai et al., 2008; Fu et al., 2012a) as follows:

$$P_{top} = (B_{hour})^2 \times (B_{size})^2 \times P_{topmax}, \tag{4}$$

$$P_{bottom} = (B_{hour})^2 \times (B_{size})^2 \times P_{bottommax}, \tag{5}$$

where $P_{topmax}$ and $P_{bottommax}$ were parameters that define the potential maximum plume heights for $P_{top}$ and $P_{bottom}$, respectively.

### 2.3 Analysis of backward trajectories and concentration-weighted trajectories

The Hybrid Single-Particle Lagrangian Integrated Trajectory version 4 (HYSPLIT4) model developed by National Oceanic and Atmospheric Administration Air Resources Laboratory (NOAA ARL) was employed to predict regional transport pathways arriving at receptor cities of interest (Stein et al., 2015). HYSPLIT4 was driven by the final global meteorology analysis data obtained from the National Centers for Environmental Prediction's Global Data Assimilation System (https://ready.arl.noaa.gov/archives.php, last access: 5 August 2019) with a $1° \times 1°$ latitude–longitude grid, and was run four times per day at starting times (i.e., 00:00, 06:00, 12:00 and 18:00 LT) with the starting height of 100 m above ground level. In this study, the 48 h back trajectories of air masses were used for further analyses. More details about the HYSPLIT4 model can be found at http://www.arl.noaa.gov/ready/open/hysplit4.html (last access: 2 August 2019).

We adopted the concentration-weighted trajectory (CWT) method to pinpoint potential major OBB sources affecting regional surface $PM_{2.5}$ concentrations during the study period based on the HYSPLIT analysis and surface $PM_{2.5}$ observations (Wang et al., 2009; Yu et al., 2014; Li et al., 2015b). The CWT method collected all concentrations of trajectories in an individual grid ($C_l$) to calculate the corresponding average CWT value ($C_{ij}$) for each grid cell (i, j) by the following Eq. (6) (Hsu et al., 2003):

$$C_{ij} = \frac{1}{\sum_{l=1}^{M} T_{ijl}} \sum_{l=1}^{M} C_l T_{ijl} \, , \tag{6}$$

where l and M denote the index and the total number of the trajectories, respectively, and $T_{ijl}$ represent the residence time for the trajectory l spent in the grid (i, j). The relatively higher CWT values imply high potential contributions to elevated $PM_{2.5}$ values at the receptor site. Thus, the weighted concentration fields can be utilized to determine the relative significance of potential sources for regional haze in CEC.

**2.4 Observational data**

Hourly mass concentrations of surface $PM_{2.5}$ and other chemical species (i.e., CO, $NO_2$, $SO_2$, $O_3$, and $PM_{10}$) were continuously measured by the Ministry of Ecological Environment of China (http://www.cnemc.cn/, last access: 2 August 2019), including 340 monitoring sites in 65 cities during the study period over CEC. The $PM_{2.5}$ compositions were obtained from the Campaign on Atmospheric Aerosol Research network of China (CARE-China) in 2011, which was mainly supported by the Chinese Academy of Sciences. The CARE-China network, as the first comprehensive measurement platform for atmospheric aerosols across China (Xin et al., 2015; Liu et al., 2018), has embraced 40 ground sites including 20 urban sites, 12 background sites, and 8 rural/suburban sites and measured most of $PM_{2.5}$ compositions (Liu et al., 2018). Table S3 displays detailed information of 5 sites utilized in this study, which were located in Hunan, Anhui, Jiangxi, Shandong, and Jiangsu. Note that, owing to their key features, including the discontinuous samples (i.e., from June 2 to 4, from June 9 to 11, and from June 16 to 18) as well as the 48h temporal resolution, they were not able to support the time series analysis but can be used for the period evaluations. These monitoring data were used as follows: (1) According to the evolution of surface $PM_{2.5}$ concentrations and their composition over CEC, we characterized changes in spatial and temporal patterns of regional haze induced by OBB. (2) We compared simulated chemical and meteorological fields with surface observations to evaluate model performance. (3) The $PM_{2.5}$ concentrations were used to estimate potential sources by the CWT method. (4) Daily mean values of AOD at 550 nm retrieved from satellite platform were examined during the target period to highlight significant spatial and temporal variabilities of regional haze over CEC. Here, the episode-averaged AOD products from MODIS (MOD08_D3) at 550 nm were utilized (https://giovanni.sci.gsfc.nasa.gov/giovanni/, last access: 5 August 2019).

To present OBB and OCSB emissions, daily fire products generated by MODIS (MCD14DL) and corresponding emissions were collected to exhibit explicit spatial and temporal evolutions of OBB. Along with the HYSPLIT4 and CWT methods, the integrated analyses of surface weather patterns from the Korea Meteorological Administration were conducted to illustrate meteorological fields triggering the regional haze over CEC.

## 3 Results and discussion

### 3.1 OBB and OCSB information in FINNv1.5

We collected monthly OCSB and other types of OBB information estimated by FINNv1.5 over CEC throughout 2014. From the perspective of the monthly variation in numbers of fire hotspots (Fig. 2), total OBB exhibited a notable seasonal pattern with one distinct peak in summer, especially in June (19 % of the yearly total value). We further examined each type of OBB classified by seven land use categories (i.e., Croplands, Grass land/savanna, Woody savanna/shrublands, Tropical forest, Temperate forest, Boreal forest, and Temperate evergreen forest) and found that the relative contribution of OCSB ranged from 13 % to 86 % and also peaked in June of 2014. This presented a highly consistent trend with that of OBB, indicating that OCSB might be the leading contributor of OBB sources over CEC. Therefore, to simultaneously investigate OCSB and other types of OBB effects over CEC, June 2014 should be the best target period on which to focus.

Figure 3 shows a remarkable variation in spatiotemporal distributions of daily OCSB and other types of OBB fire hotspots over CEC from June 1 to 19, exhibiting a complete OBB evolutionary process. An OBB outbreak event occurred from June 5 to 14 with the maximum number of fire hotspots (more than 800). By contrast, there existed much fewer fire hotspots in the other periods. This period can, in turn, be divided into the following three episodes: EP1 (from June 1 to 4), EP2 (from June 5 to 14) and EP3 (from June 15 to 19), which represented the pre-OBB, OBB and post-OBB stages, respectively. It should be noted that OCSB makes uniformly much larger relative contributions to total OBB than other types in terms of the number of fire hotspots for the study period with more than 80 % during EP2.

During EP1 and EP3, limited OBB fire hotspots were sparsely distributed over CEC, as shown in Fig. 3. For the period of the OBB outbreak (EP2), most of the OCSB fire hotspots concentrated in Henan (32 %) and Anhui (41 %), where local OCSB emissions might play a key role in shaping spatiotemporal distributions of $PM_{2.5}$ concentrations. Owing to regional transport, surrounding areas, such as Hubei, Hunan, and Jiangxi, might be significantly influenced by OCSB, depending on meteorological conditions. The activity levels of other types of OBB during EP2 also increased by 250 % compared to normal, particularly in Anhui, Zhejiang, and Jiangxi, although it was still significantly lower than those of OCSB, as indicated in Fig. 3.

Table S4 summarizes original provincial emissions of gaseous and particulate species for OCSB and total OBB during the study period based on the FINNv1.5 emission estimations. Most (85 %) OBB emissions occurred during EP2. As expected, OCSB dominated OBB emissions (74 % ~ 94 %) by producing 3040, 17, 10781, 367, and 399 million moles of NMVOCs, $SO_2$, CO, $NH_3$, and $NO_X$, respectively, and 1877, 8977, 15778, 19097 tons of EC, OC, primary $PM_{2.5}$ and $PM_{10}$, respectively. During EP2, Henan and Anhui were the top two contributors over CEC and accounted for 66 ~ 76 % of OBB emissions. Specifically, from the perspective of total OBB emissions in Henan and Anhui, OCSB emissions in these two provinces contributed 94 % of NMVOCs, 72 % of $SO_2$, 87 % of CO, 88 % of $NH_3$, 90 % of $NO_X$, 89 % of EC, 76 % of OC, 74 % of $PM_{2.5}$ and 75 % of $PM_{10}$, respectively. Other types of OBB emissions occurred mainly in Anhui (29 %), Zhejiang (28 %) and Jiangxi (14 %). Nevertheless, OBB emissions associated mainly with OCSB estimated in the original FINNv1.5 were projected

to be substantially underestimated due to a variety of satellite-based limitations, such as unresolved small and ephemeral agricultural fires, occlusion effects of prevalent cloud cover, or discontinuous time spans (Streets et al., 2003;Wiedinmyer et al., 2011; Randerson et al., 2012; Zhou et al., 2018; Uranishi et al., 2019) .

### 3.2 Characteristics of observed regional haze pollution

Both ground-level and space observations were used to diagnose this OBB event and its associated regional haze (Figs. 4, 5, and 6). Figure 4 shows the magnitudes and spatial distributions of observed episode-averaged $PM_{2.5}$ concentrations over CEC for EP1, EP2, and EP3. Before the OBB outbreak (EP1), there were no extreme $PM_{2.5}$ concentrations, all of which were generally lower than 75 μg m$^{-3}$. For EP2, elevated $PM_{2.5}$ concentrations (> 115 μg m$^{-3}$) were observed in Henan and Anhui. Apparently, this was highly consistent with spatiotemporal distributions of OCSB (Fig. 3), which was thus anticipated to cause potential prominent impacts on local haze formation. On the other hand, regional haze also occurred in Hubei and Hunan, where no extensive high-intensity OCSB emissions were observed during the study period (Fig. 3). Thus, not only OCSB emissions but also regional transport played a key role in regional haze. It should be noted that limited emissions from other types of OBB also have a certain impact. With the sudden decline in OBB and $PM_{2.5}$ concentrations over CEC during EP3, however, moderate regional haze ($PM_{2.5}$ concentrations >75 μg m$^{-3}$) lingered in Henan and Anhui. In addition to anthropogenic contributions, this phenomenon is probably a result of unfavorable meteorological conditions, which trapped previously generated emissions during EP2.

Figure 5 shows spatial distributions of episode-averaged AOD observed by MODIS (MOD08_D3) at 550 nm during EP2. It is in good agreement with spatial distributions of surface average $PM_{2.5}$ concentrations. For instance, much higher AOD values were mostly detected in Henan, Anhui, Hubei, and Hunan, associated with relatively high surface observed $PM_{2.5}$ concentrations and substantial OCSB emissions, as shown in Figs. 3 and 4. In addition, the satellite-based product detected that spatial distributions of high AOD values covered wider areas than the surface measurements, such as in Jiangxi, Zhejiang and Fujian. This was possibly due to the fact that PM suspended in the upper troposphere was more easily transported than that on the ground. This phenomenon further illustrates that OBB dominated by OCSB is not only a significant local source but also an important regional source.

Collectively, seven provinces (i.e., Henan, Anhui, Hubei, Hunan, Jiangxi, Zhejiang, and Fujian) are the focal areas affected by OBB. To further understand the evolution of regional haze characteristics, we focus on these seven provinces to analyze time series of provincial hourly $PM_{2.5}$ concentrations during the study period (Fig. 6). Correspondingly, the observed trends were also divided into three distinct stages on the basis of $PM_{2.5}$ concentrations. For EP1, hourly $PM_{2.5}$ concentrations over CEC were relatively low and almost never exceeded 75 μg m$^{-3}$. Specifically, the average values in these seven provinces during this episode were 50 μg m$^{-3}$, 59 μg m$^{-3}$, 71 μg m$^{-3}$, 65 μg m$^{-3}$, 44 μg m$^{-3}$, 46 μg m$^{-3}$, and 41 μg m$^{-3}$, respectively, depicting a period with a relatively clean air environment. For EP2 with the OBB outbreak mainly associated with OCSB, $PM_{2.5}$ concentrations increased significantly in most of these provinces, and the mean $PM_{2.5}$ concentrations in these seven provinces reached values of 135 μg m$^{-3}$, 175 μg m$^{-3}$, 182 μg m$^{-3}$, 147 μg m$^{-3}$, 129 μg m$^{-3}$, 72 μg m$^{-3}$, and 37 μg m$^{-3}$, respectively. This indicates the

extensive haze in Henan, Anhui, Hubei and Henan. For Jiangxi and Zhejiang during EP2, no severe haze was detected despite large increases in $PM_{2.5}$ concentrations relative to those in EP1. Thereafter, $PM_{2.5}$ concentrations in most parts of CEC decline rapidly and gradually stabilize in a lower range during EP3. Overall, the spatiotemporal evolution trends of regional haze characteristics were highly related to those of OBB emissions as analyzed above, signifying the dominant role of OBB
emissions. As shown in these results, two important issues can be identified. (1) The temporal variations in $PM_{2.5}$ concentrations in Hubei were highly similar to those in Hunan during the target period, especially in EP2, revealing that coincident unfavourable meteorological fields might facilitate severe haze there. (2) Relative to the evolution of regional haze in Henan, there was a distinct time lag (24 h ~ 48 h) for those in Hubei and Hunan. This indicates that OBB emissions mainly associated with OCSB in Henan as well as subsequent regional transport might also be responsible for severe haze in Hubei
and Hunan.

Besides the analysis of $PM_{2.5}$ concentrations at the provincial levels, the major $PM_{2.5}$ compositions (i.e., $K^+$, $SO_4^{2-}$, $NO_3^-$, OC, and $NH_4^+$) at the individual sites (Table S5) were also illustrated. We found that there were steady increases in most of the chemical compositions during EP2 at the Changsha and Yucheng sites, which were located in Hunan and adjacent to Anhui, respectively. Such spatiotemporal patterns were highly correlated with those of OBB and $PM_{2.5}$ concentrations. The $SO_4^{2-}$ and
335 OC, as the dominating species, were project to be responsible for such increases during EP2. For example, the former rose by 13.4 μg m$^{-3}$ (65.2 %) and 17.6 μg m$^{-3}$ (134.5 %) in Changsha and Yucheng, respectively. More, we should pay special attention to $K^+$, which was usually treated as the tracer for OBB (Duan et al., 2004). As expected, with the OBB outbreaks in EP2, the $K^+$ concentrations dramatically increased at those two sits (25.2 % ~ 153.7 %), indicating the critical role of OBB in Hunan and Anhui. A similar phenomenon also appeared in Qianyanzhou, where the observed $K^+$ concentrations increased from 0.09
340 μg/m$^3$ to 0.56 μg/m$^3$ (534.2 %). Note that, most of other observed composition in Qianyanzhou and Wuxi remained in the relatively low range.

In conclusion, from the point of view of the highly correlated relationship between regional haze and OBB activities, we can infer that OBB mainly associated with OCSB might dominate spatial and temporal variations in relatively high $PM_{2.5}$ concentrations over CEC, especially in Henan, Anhui, Hubei, and Henan. Besides, meteorological fields should also play a
345 critical role in shaping spatiotemporal variations in the regional haze, whereas other types of OBB emissions might be responsible for local haze in Zhejiang and Jiangxi. Therefore, relative contributions of OCSB and other types of OBB, as well as relevant meteorological fields, to regional haze formation at different regional scales are not consistent and should be accurately examined.

**3.3 Constrained optimal OBB emissions**

Given large uncertainties in the OBB emissions estimated by the original FINNv1.5, it is necessary to refine these emissions in order to allow the WRF-CMAQ model to reproduce the magnitudes and spatiotemporal distributions of regional haze induced by this OBB event. Also, it is the prerequisite to accurate estimations of OCSB and other types of OBB contributions. In the present study, all available observations from 340 ground monitoring sites and the WRF-CMAQ model simulations

were used in co-constraining OCSB and other types of OBB emissions from FINNv1.5. This constraining method comprised
two steps: (i) to characterize the nonlinearity between emissions and $PM_{2.5}$ concentrations, we examined model responses to
variable OBB emission perturbations, which referred to amplifying OBB emissions using coefficients arranged in an arithmetic
sequence (Xiao et al., 2010; Digar and Cohan, 2010; Tang et al., 2011), the first term of which = 1, consistent with the constant
difference of terms; (ii) subsequently simulated $PM_{2.5}$ concentrations were then compared with available observations. Hence,
with variable emission perturbations, this evaluation process would be repeated, and simulated results could gradually
approach observations. During the evaluation processes, the normalized mean bias (NMB) was selected as the major indicator
to characterize the discrepancy (Yu et al., 2006).

The constraining method took consideration of the spatiotemporal heterogeneity of uncertainties in OBB emissions to derive
the optimal adjustment coefficients for OBB emissions more accurately. According to the variations in the OBB activity level,
as shown in Fig. 3 and Table S4, the entire study period was divided into three periods, as noted above, i.e., the pre-OBB (EP1),
OBB (EP2) and post-OBB (EP3) stages. Nevertheless, in terms of regional haze, only EP1 was treated as the normal situation
under the consideration that $PM_{2.5}$ concentrations during EP3 would be inevitably impacted by residual emissions produced
by this OBB event during EP2. To eliminate potential uncertainties in other emission sources (i.e., anthropogenic and biogenic
emissions), we evaluated the model capability under normal situations (no OBB event), that is, the model was driven by
original OBB emissions as well as anthropogenic and biogenic emissions (referred to as the BASE case). Figure 6 illustrates
the simulated provincial average $PM_{2.5}$ concentrations in the BASE case during the study period for Henan, Anhui, Hubei,
Hunan, Jiangxi, Zhejiang, and Fujian. The results show that the observations during the study period were uniformly
underestimated. However, the NMBs were within a reasonable range (> -25%) during EP1 but presented much lower values
(> 40 %) during the rest periods. Thus, this demonstrates that original OBB emissions as well as other sources can allow the
model to capture spatiotemporal variations in $PM_{2.5}$ concentrations for normal situations without significant impacts of OBB
in most of provinces, as recorded by previous studies (Hu et al., 2015; Liu et al., 2016; Qiao et al., 2019), whereas the model
failed to reproduce the rapid outbreak of $PM_{2.5}$ concentrations for this OBB event, especially during EP2 in Henan and Anhui.
This result shows the need to derive optimal OBB emissions using this constraining method. On the other hand, for EP1, we
also need to overcome slight underestimations of simulated $PM_{2.5}$ concentrations in order to improve model performance and
subsequent estimates of OBB contributions.

Based on the comprehensive analyses for OBB emissions and their associated regional pollution, our goal here is to extend
this constraining method to specific spatiotemporal scales. The three study stages (EP1 ~ EP3) should be discriminated due to
their distinct OBB emissions, namely, different coefficients should be adopted to adjust OBB emissions for each individual
study period. As discussed in Sections 3.1 and 3.2, Henan and Anhui were the major contributions to local OBB emissions (>
80 %) during the entire study period and would affect other provinces through regional transport. Hence three areas (Henan,
Anhui, and other provinces over CEC) were separately adjusted as shown in Fig.7. To do so, we need to employ this
constraining method in the following ways: (i) derive respective optimal OBB emissions in Henan and Anhui for EP1 at first
and then for EP2. Note that the optimal adjustment coefficient for EP1 would also be appropriate for EP3 due to the fact that

both of them represent the normal situation with respect to the OBB emissions; (ii) on the basis of those results, the next step was to derive corresponding optimal OBB emissions for other provinces.

Figure 7 exhibits colored cells to represent NMBs between observed and simulated $PM_{2.5}$ concentrations that characterize model responses to variable OBB emission perturbations in Henan, Anhui, and other provinces over CEC. The numbers on the X and Y axes refer to the adjustment coefficients for OBB emissions for EP1 and EP2, respectively. With variable OBB emission perturbations, simulated results gradually approached to observed values. For Henan and Anhui, OBB emissions with relatively low adjustment coefficients (1 ~ 5) do not allow the model to capture observed high $PM_{2.5}$ concentrations during

EP2, while those with higher adjustment factors (> 4) would cause the model to significantly overestimate observed values during EP1. This highlights the substantial underestimation of original OBB emissions in FINNv1.5, especially during the OBB outbreak period (i.e., EP2). By comparison, relatively low coefficients should be adopted to adjust OBB emissions in other provinces. These results point to the importance of variable emission perturbations by considering the spatiotemporal heterogeneity in optimizing OBB emissions. Specifically, we determine that optimal coefficients for OBB emissions for Henan,

Anhui, and other provinces are 6, 7, and 4 for EP2, respectively, and 4, 4, and 2 for both EP1 and EP3, respectively. Correspondingly, all of the NMB values reached the minimum and were controlled within ±5 %. In summary, we employ the aforementioned optimal coefficients to adjust OBB emissions at specific spatiotemporal scales. Table 1 summarizes the spatiotemporal emission amounts for all the species (NMVOCs, $SO_2$, CO, $NH_3$, $NO_X$, EC, OC, $PM_{2.5}$, $PM_{10}$) for each province over CEC for each stage of the study period (EP1 ~ EP3). Corresponding maps of $PM_{2.5}$ emissions for OCSB and other types of OBB presented similar spatial distributions to those of fire hotspots as shown in Fig. 8. By taking $PM_{2.5}$ emitted from OCSB

as an example, emission intensities in some pixels were found to exceed a value of 100 tons during the study period, and most cases were found in the areas surrounding the common border of Henan and Anhui. For other provinces, $PM_{2.5}$ emission intensity only ranged from 1 to 20 tons per pixel. Other species from OCSB emissions presented similar spatial distributions. On the other hand, all species for other types of OBB consistently showed sparse spatial distributions and limited emissions.

In summary, optimal OBB emissions were derived by the constraining method which integrated observations and model results based on FINNv1.5. Note that this constraint was established based on specific spatial scales separately (i.e., Henan, Anhui, and rest provinces over CEC).

## 3.4 Model evaluation

Accurate simulations of meteorological and chemical fields were prerequisite to precise estimations of OBB contributions to

the haze formation. Few previous OBB emissions established by bottom-up or top-down methods were evaluated directly. Here we assessed the optimal OBB emissions with the WRF-CMAQ model. This case is abbreviated as the OPT case hereinafter. The simulated chemical species (i.e., $PM_{2.5}$, $O_3$, $SO_2$, CO, and $NO_2$) and meteorological parameters (i.e., 10 m wind speed and direction, 2 m temperature, and PBLH) from the model were both taken into considerations by comparing with ground observations.

Figure 4 illustrates the spatial distributions of observed and simulated hourly PM$_{2.5}$ concentrations in the OPT case for the entire CEC in EP1, EP2, and EP3. As discussed in Section 3.3, the OBB emissions from FINNv1.5 can drive the model to achieve reasonable results for typical situations, such as EP1, which, however, remained as slight underestimations of observations. It is important to note that the simulated PM$_{2.5}$ concentrations in the BASE case were significantly lower than observations at most of monitoring sites during EP2, especially in Henan, Anhui, Hubei, and Hunan as noted above. This

implies that the original OBB emissions from FINNv1.5 substantially underestimate the emissions during the period of the OBB outbreak (i.e., EP2). Compared to the BASE case, the model results in the OPT case were able to reproduce more accurately the magnitudes and spatial patterns of observed PM$_{2.5}$ concentrations. Specifically, the model with the optimal OBB emissions not only captures extremely high observations (> 115 μg m$^{-3}$) in Henan, Anhui, Hubei, and Hunan for the OBB event during EP2, but also succeeds in reproducing relatively lower values in other provinces. For EP1 and EP3, simulated

results in the OPT case increased to some extent as expected and became closer to observations by comparison. Besides, compared with the satellite retrievals, the model-derived AODs in the OPT case during EP2 presented the extremely similar spatial patterns over CEC (Fig. 5). Especially, they could reproduce the relatively high measurements over Henan, Anhui, Hubei, and Hunan. Nevertheless, we recognized the general underestimations of model-derived AODs, in particular over the areas with the extremely PM$_{2.5}$ concentrations, which might be duo to the uncertainties in the numerical predictions of the

plume rise of OBB (Tai et al., 2008; Fu et al., 2012a). Another explanation may be contamination of the observed AODs due to opaque clouds as described by several studies (Huang et al., 2012; Aouizerats et al., 2015). These results establish reliable model performance.

To further evaluate the model performance, we examined the model capabilities at the provincial scale. Figure 6 displays the comparisons of observed and simulated hourly PM$_{2.5}$ concentrations for the OPT case in Henan, Anhui, Hubei, Hunan, Jiangxi,

Zhejiang, and Fujian. Besides the NMB, the correlation coefficient (R) was also used to evaluate the simulated results. Generally, the WRF-CMAQ model with the optimal OBB emissions can reasonably capture the temporal evolution of surface PM$_{2.5}$ concentrations for these seven provinces. With the abundant observations as the constraint, the NMB values in these seven provinces were 1.1 %, 3.3 %, 0.2 %, 1.3 %, 1.4 %, 6.2 %, and 22.8 %, respectively, with R values of 0.90, 0.80, 0.53, 0.74, 0.56, 0.80 and 0.64, respectively. These results provide direct support for the reliable capability of the optimal OBB

emissions, reproducing the steep rising curves of observed PM$_{2.5}$ concentrations in Henan, Anhui, Hubei, and Hunan during EP2 when OBB might trigger regional haze over the central part of CEC, as shown in Figs. 4 and 6.

The first two peaks of observed PM$_{2.5}$ during the study period occurred in Anhui and Henan. Observed PM$_{2.5}$ concentrations in Anhui and Henan gradually increased from 42 μg m$^{-3}$ on June 1 to 226 μg m$^{-3}$ on June 7 and from 75 μg m$^{-3}$ on June 7 to 226 μg m$^{-3}$ on June 10, respectively. Correspondingly, the model in Anhui and Henan captures this general temporal pattern,

especially the maximum with NMB values of 9 % and 12 %, respectively. In contrast, regional transport might play a critical role in dominating two sudden outbreaks of observed PM$_{2.5}$ concentrations in Hubei and Hunan with relatively high values of 335 μg m$^{-3}$ and 252 μg m$^{-3}$ on June 12, respectively. The simulations to a large extent capture these two abrupt peaks, even though consistently underestimating the maxima with the NMB values ranging from 20 % to -15 %. Note that a distinct 12 h

time lag exists between the observed and simulated peaks of $PM_{2.5}$ concentrations in Hubei, which is associated with possible
uncertainties in meteorological simulations.

Besides, we also evaluated the simulated $PM_{2.5}$ composition ($K^+$, $SO_4^{2-}$, $NO_3^-$, OC, and $NH_4^+$) (Table S5). Compared to the BASE case, the simulated results in the OPT case at the Changsha, Hefei, and Yucheng sites were raised in various degrees and improved in general. Specifically, the NMB values of the simulated $SO_4^{2-}$ and $NH_4^+$ were reduced from – 47.5 % to -14.1 % and from -30.4 % to -0.9 %, respectively, marking the reliable role of the constraining method to a large extent. The simulated
OC concentrations presented similar growth trends but remained inadequate (-48.7 % ~ -6.6 %), indicating the missing mechanisms and insufficient simulations of secondary organic aerosols in the AERO6 scheme. On the contrary, the OPT case was prone to underestimate $K^+$ (-79.1 % ~ -20.0 %) and fail to capture the spatiotemporal changes in $NO_3^-$ (-69.3 % ~ 391.0 %). This might be associated with the large uncertainties in the chemical speciation in the original OBB emissions. However, it should be noted that the OPT case can effectively stabilize the model performance in reproducing $NO_3^-$ (~ 24.7 %) when their
values were larger than 10 μg m$^{-3}$. In addition, the constrained OBB emissions also improved the model performance at the Qianyanzhou site, while the Wuxi site seemed to be irrelevant to the OBB case. Hence, the constraining method enables the model to optimize the simulated $PM_{2.5}$ composition over CEC to some extent.

For other chemical species and main meteorological parameters, Tables S6 and S7 indicate that the WRF-CMAQ model with the optimal OBB emissions also reproduces reasonable results, with most R values greater than 0.7 and NMB values within ±
30 %. Specifically, compared to the BASE case, the NMB and R values for the species (i.e., CO, $PM_{10}$, and $PM_{2.5}$) largely contributed by OBB emissions in the OPT case were also considerably improved, reaching a level of ± 25 %.

### 3.5 Relative contributions of OCSB and OBB to regional haze formation

We designed three model simulations to simultaneously isolate OCSB and OBB contributions to $PM_{2.5}$ concentrations over CEC, i.e., the first with only anthropogenic and biogenic emissions (referred to as the NOBB case), the second with not only
anthropogenic and biogenic emissions but also constrained OCSB emissions (referred to as the OPT_OCSB case), and the last with not only anthropogenic and biogenic emissions but also total constrained OBB emissions (namely the OPT case). In this study, OCSB and OBB contributions were quantified by differences between simulated $PM_{2.5}$ concentrations in different cases in terms of absolute mass concentrations and relative contributions:

$$OCSB_{mass} = OPT\_OCSB_{mass} - NOBB_{mass}, \tag{7}$$

$$OCSB_{contribution} = (OCSB_{mass} / NOBB_{mass}) \times 100\%, \tag{8}$$

$$OBB_{mass} = OPT_{mass} - NOBB_{mass}, \tag{9}$$

$$OBB_{contribution} = (OBB_{mass} / NOBB_{mass}) \times 100\%, \tag{10}$$

where $NOBB_{mass}$, $OPT\_OCSB_{mass}$ and $OPT_{mass}$ represent simulated $PM_{2.5}$ concentrations in the BASE, OPT_OCSB, and OPT cases, respectively. $OCSB_{mass}$ ($OBB_{mass}$) and $OCSB_{contribution}$ ($OBB_{contribution}$) denote contributions from OCSB (OBB)
with regard to absolute $PM_{2.5}$ mass concentrations and relative contributions, respectively.

Figure 9 shows spatiotemporal distributions of $OBB_{mass}$ and $OBB_{contribution}$ during EP1, EP2, and EP3. For CEC, OBB during EP2 had much more significant impacts than those during EP1 and EP3 on regional $PM_{2.5}$ concentrations. In particular, OBB contributed up to more than 75 % (> 75 µg m$^{-3}$) to surface $PM_{2.5}$ concentrations over the border of Henan and Anhui during EP2. These spatial distributions were exactly consistent with those of OBB emissions and the regional haze, implying the key role of OBB emissions in triggering the regional haze formation. In contrast, $PM_{2.5}$ concentrations also increased substantially by 14 ~ 74 µg m$^{-3}$ (16 ~ 78 %) in Hubei and Hunan, where no apparent OBB emissions were detected. This phenomenon should be attributed to regional transport of air pollution. It is worth noting that, there were the substantial declines in OBB emissions and $PM_{2.5}$ concentrations with significant decreases in OBB contributions consistently occurring over CEC during EP3. This indicates that stringent controls on the OBB emissions are still conspicuously beneficial for regional air quality. On the other hand, massive OCSB pollutants mainly emitted in Henan and Anhui were not transported to the eastern coastal zones of CEC during EP3. In addition, local OBB would not lead to regional haze in the southern edge of CEC despite its large contributions (> 75 %). As shown in Figure 10, OCSB contributions show largely similar spatiotemporal distributions to those of OBB, indicating that OCSB generally played a key role in reshaping $PM_{2.5}$ concentrations over CEC. During the OCSB outbreak period, its relative contributions were significantly higher than those in other periods. However, OCSB contributions were located at some typical areas, namely the common border between Henan and Anhui (> 75 µg m$^{-3}$, 75 %) and central narrow zones between Hubei and Hunan (> 35 µg m$^{-3}$, 39 %). For other provinces, its relative contributions were within a narrow range of approximately 20 %. Figure 11 highlights that fairly large proportions (more than 80 %) of OBB contributions came from OCSB over the most parts of CEC, indicating that OCSB should play a dominant role in reshaping spatial and temporal distributions of regional $PM_{2.5}$ concentrations over the most parts of CEC during the study period, especially for EP2. In contrast, other types of OBB barely influenced the haze formation in Jiangxi, Fujian, and Zhejiang (< 10 µg m$^{-3}$).

To further understand OCSB and OBB effects on regional haze formation over CEC, time series of OBB and OCSB contributions to hourly $PM_{2.5}$ concentrations for seven provinces over CEC during the study period are examined in Fig. 6. Overall, OBB emissions had significant contributions to hourly $PM_{2.5}$ concentrations in Henan (44.14 %, 23.82 µg m$^{-3}$), Anhui (46.19 %, 36.24 µg m$^{-3}$), Hubei (56.89 %, 37.69 µg m$^{-3}$), Zhejiang (25.89 %, 17.79 µg m$^{-3}$), and Fujian (28.97 %, 7.31 µg m$^{-3}$), but lower impacts on those in Hunan (21.51 %, 13.86 µg m$^{-3}$) and Jiangxi (14.07 %, 7.11 µg m$^{-3}$). It should be noted that most of OBB (> 89 %) contributions uniformly resulted from OCSB emissions. For the OBB outbreak period, significant OBB contributions were found in Henan (60.47 %), Anhui (57.73 %), Hubei (42.91 %), and Hunan (29.14 %), whereas those in other provinces were in a small range (± 15 %), and OCSB emissions always contributed to the most (more than 90 %). Note that the variations in $OPT\_OCSB_{mass}$ in Henan and Anhui kept generally synchronized with those in local abundant OCSB emissions and haze. In contrast, the peak values of $OPT\_OCSB_{mass}$ in Hubei and Hunan, where no extensive fire hotspots were detected, showed apparent temporal asynchronies (24h ~ 48h) relative to those in Henan. This means OCSB was not only one of the significant local pollution sources but also a considerable regional source over CEC. On the other hand, other types of OBB emissions had an impact on local $PM_{2.5}$ concentrations in Jiangxi (9.04 %), Fujian (11.82 %), and Zhejiang (10.08 %) but nominal effects (< 5 %) on the other provinces.

**3.6 Potential OCSB emission sources and associated meteorological causes**

OCSB emissions can have significant impacts on local and regional haze in Henan, Anhui, Hubei, and Hunan. There is thus a need to explore associated potential sources and meteorological conditions. First, we utilized the CWT method to investigate potential OCSB emission sources triggering regional severe haze. To pinpoint potential emission sources for OCSB, Pingdingshan, Hefei, Wuhan, and Changsha were selected as receptor sites to represent Henan, Anhui, Hubei, and Hunan, respectively. Figure 12 presents spatial distributions of the CWT values calculated based on the hybrid receptor model (Fig. S1) and observed $PM_{2.5}$ concentrations for these cities. It should be noted that only sources with relatively high CWT values ($> 150\ \mu gm^{-3}$) were used as potential major sources leading to regional severe haze. For Henan and Anhui, most of the potential major sources concentrated in local areas with large amounts of OCSB emissions during EP2, thereby suggesting the dominant role of local OCSB emissions. For Hubei and Hunan, however, potential major sources were also located in common areas in Henan and Anhui, especially over the border between them, thus clearly demonstrating that local haze in Hubei and Hunan was caused by regional transport of emissions from Henan and Anhui. In summary, OCSB emissions in Henan and Anhui should be mainly responsible for regional haze over CEC during EP2.

Atmospheric pollution processes, especially wind fields, are closely related to synoptic weather patterns. A high-pressure system is usually considered to be the governing synoptic pattern contributing to accumulation of air pollutants, whereas the differences between high and low-pressure systems are thought to be the major cause of transport on the regional scale. Figure 13 uses surface pressure maps from June 8 to 11 to explore the relationships among synoptic pressure patterns, OBB emissions, and haze formation over CEC. On June 8, concurrent with the OCSB outbreak, an extensive high-pressure system enveloped over CEC, forming a stationary condition and facilitating rapid accumulations of $PM_{2.5}$ in Henan and Anhui. Thereafter, it was gradually turned into the low-pressure system on June 9 located at Jiangxi, receiving pollutants originating from Henan and Anhui by the differential field between the high- and low- pressure systems. This differential field also established positive dispersion conditions for Anhui, although OCSB emissions were still occurring there. Thereafter the stationary high-pressure system made a brief stay in Henan and Anhui on June 10, being prone to accumulate elevated $PM_{2.5}$ concentration, then a narrow low-pressure corridor squeezed by two high-pressure zones went through Henan, Hubei, and Hunan, thus facilitating regional transport of severe haze.

**3.7 Comparison to previous studies**

Numerous emission data on OBB have been established for CEC. These results, generally estimated based on statistics or satellite retrievals, have not generally been constrained by model and observations. Here we compare our constrained results with five representative emission datasets including GFASv1.0 (Kaiser et al., 2012), GFEDv4.1s (Van Der Werf et al., 2017), FINNv1.5 (Wiedinmyer et al., 2011), Qiu et al. (2016), and Zhou et al. (2017), as shown in Fig. 14. The former four inventories belong to satellite-based products, while the last one is derived with the traditional bottom-up method. Theoretically, there are two additional issues we need to note: (i) GFEDv4.1 incorporates small fires through the MODIS active fire detection product

(Randerson et al., 2012); (ii) Qiu et al. (2016) is unable to discriminate small fires, but it can also detect more fire hotspots than GFASv1.0 and FINNv1.5 with the benefit of the extra MODIS burned area product (MCD64Al).

Figure 14 shows that the constrained OBB emissions in this study were generally higher than satellite-based estimates but
lower than the bottom-up statistics (i.e., Zhou et al., 2017). For example, the constrained CO emissions over CEC were 5.2, 3.4, and 5.7 times larger than those in GFASv1.0, GFEDv4.1s, and FINNv1.5, respectively. By comparison, the constrained $SO_2$ and $NO_X$ emissions were also closer to those of GFED v4.1s. This indicates that small fires were projected to be central to OBB emissions since this OBB event was dominated by OCSB as mentioned, which always come from individual farmers with limited burned areas. In turn, for the NMVOCs, EC, OC, $PM_{2.5}$, and $PM_{10}$, our estimates were much higher than those
from GFEDv4.1s but closer to those derived from Qiu et al. (2016) and Zhou et al. (2017) with the difference ranging from - 10 % to 73 %. Thus, relative to substantial underestimations in GFEDv4.1s despite of additional small fires, such large increases in Qiu et al. (2016), which also acted as satellite-based estimates, should be attributed to extra detected fire hotspots. An emphasis should be given to Zhou et al. (2017), especially on higher $SO_2$, CO, $NO_X$, and $NH_3$ emissions (> 4 times), inevitably leading to unreasonable overestimations of $PM_{2.5}$ concentrations. On the other hand, it involves various uncertainties
due to inherent statistical parameters and has common disadvantages in allocating seemingly sufficient emission amounts into high spatiotemporal cases.

In summary, the constrained OBB emissions in this study can not only supplement satellite-based estimates, but also to a large extent avoid overestimations due to inherently uncertainties originating from bottom-up statistics.

In contrast to numerous OBB emission inventories for CEC, few studies focus on extreme OBB events over CEC to investigate
OBB contributions to surface $PM_{2.5}$. Cheng et al. (2014) applied numerical models with satellite-based OBB emissions and found obvious OBB effects (i.e., 23 % ~ 48 %, 18 µg m$^{-3}$ ~ 65 µg m$^{-3}$) on surface $PM_{2.5}$ in several cities over CEC (i.e., Ningbo, Shanghai, Nanjing, Hangzhou, and Suzhou) in June 2011. During the same period, Wu et al. (2017) utilized similar methods and found similar results in Nanjing (50 %). As expected, each of the two studies with raw satellite-based OBB substantially underestimated surface $PM_{2.5}$ concentrations (-7 % ~ -38 %), especially in areas with intensive OBB activities at several times
with peak concentrations (< -200 %). This would thus lead to relatively low estimates of OBB effects compared with our results. A most recent study (Yang and Zhao, 2019) adopted the constraining method, that is, similar to that in this study, to optimize OBB emissions for June 2012 but ignored the spatiotemporal heterogeneity of their adjustments, resulting in much more significant estimates of OBB effects (50 % ~ 70 %) than before, while slightly lower than those in our study. This implies that more detailed constrained methods, such as provinces-specific, even grids-specific adjustments, should be helpful
to establish reliable OBB emissions with high spatial and temporal resolution over CEC.

## 4 Conclusion and discussion

OBB, especially OCSB, has long been suspected of being the source of rapid deterioration of regional air quality over CEC during the harvest seasons. Up to now, satellite-based FINNv1.5 provides a unique opportunity to constrain variations in OBB and OCSB emissions with a high spatiotemporal resolution, despite substantial underestimations mainly resulting from

unresolved agricultural small fires surrounding large fire hotspots. Here, we selected June 2014 as the study period, which includes a complete spatiotemporal evolution process (i.e., from June 1 to 19) of OBB and regional haze over CEC. We combined ground $PM_{2.5}$ measurements and model predictions with FINNv1.5 to constrain OBB emissions in terms of spatiotemporal variations. It is demonstrated that the constrained optimal emissions can allow the WRF-CMAQ to reproduce spatiotemporal chemical fields induced by OBB more reasonably. By comparison, the optimal OBB emissions can, to a large

extent, not only supplement insufficient estimations derived from satellite retrievals but also reduce overestimates of bottom-up methods. These model results were thus used to simultaneously isolate OBB and OCSB impacts on spatial and temporal evolutions of $PM_{2.5}$ concentrations. Further, we employed the CWT method as well as analysis of surface pressure maps to explore potential major OBB sources and corresponding meteorological contributions. These results can provide an effective and efficient reference for policymakers to improve environmental control strategies. The results are summarized as follows:

1. OCSB dominated OBB emissions during the study period by accounting for 74 ~ 94 %, with 81 ~ 88 % of OBB emissions occurring during EP2 (from June 5 to 14). OBB mainly associated with OCSB emissions presented significant spatial and temporal inhomogeneities, mainly (> 60 %) concentrating in Henan and Anhui during EP2. Meanwhile, a strong spatiotemporal correlation with local haze occurred in Henan and Anhui, indicating its critical role in affecting local $PM_{2.5}$ concentrations. In addition, they might also trigger regional haze in Hubei and Hunan through regional transport. By

comparison, most of other types of OBB emission (> 71 %) were located in Anhui, Zhejiang, and Jiangxi.

2. The optimal adjustment coefficients for OBB emissions over CEC during the study period were not constant but varied depending on their spatial and temporal evolutions. Original FINNv1.5 roughly underestimated OBB emissions by a factor of 5 ~ 7 during EP2, but only a factor of 2 ~ 4 during EP1 and EP3. Specifically, with respect to Henan, Anhui, and other provinces over CEC, the optimal adjustment factors of OBB emissions were 6, 7, and 5 for EP2, respectively, whereas they were 4, 4,

and 2 for EP1 and EP3, respectively.

3. With constrained optimal OBB emissions, the WRF-CMAQ model could reproduce chemical and meteorological fields reasonably during the study period from regional to provincial scales. Specifically, the model captured the rapid outbreak of $PM_{2.5}$ concentrations in Henan, Anhui, Hubei, and Henan. The correlation coefficients (R) between simulated and measured $PM_{2.5}$ concentrations succeeded 0.6, and most of the corresponding NMB values were within 10 % over CEC.

4. OBB played a key role in reshaping spatial and temporal distributions of regional $PM_{2.5}$ concentrations over CEC during the study period, and up to more than 89 % contributed by OCSB, especially in Henan, Anhui, Hubei, and Hunan during EP2. By comparison, other types of OBB also to a certain extent influenced the haze formation in Jiangxi, Fujian, and Zhejiang.

5. OCSB was not only a critical local source but also had substantial impacts on regional haze. Potential major OCSB emission sources leading to severe haze in Henan, Anhui, Hubei, and Hunan during EP2 were mostly located in Henan and Anhui,

especially their borders. This finding highlights that effective and efficient OCSB control strategies were not implemented in these regions. Conversely, other types of OBBs only exhibited their impacts on local haze in Anhui, Zhejiang, and Jiangxi.

6. Stationary high-pressure systems led to relatively stable conditions in Henan and Anhui and thus played a cardinal role in enhancing local $PM_{2.5}$ concentrations during EP2. Moreover, the transitions between the high- and low-pressure systems drive

regional transport, directly facilitating accumulation of pollutants from Henan and Anhui to Hubei, Hunan, and Jiangxi.
Therefore, interprovincial joint enforcement actions in terms of OBB prohibitions should be strictly undertaken.

To optimize the OBB emissions, previous attempts always focused on the primary emissions but neglected aerosol-radiation interactions (Uranishi et al., 2019; Yang and Zhao, 2019). Theoretically, with the aid of the systematic considerations of aerosol-radiation interactions in the two-way coupled WRF-CMAQ model, this study was projected to achieve more reliable results. A lot of previous studies have applied the same or similar model to explore the impacts of aerosol-radiation interactions.

However, it has been found that this process generally enhanced surface $PM_{2.5}$ concentrations by 8 ~ 12 % over CEC (Wang et al., 2014, 2015; Zhang et al., 2015, 2018; Jung et al., 2019). Compared to the significant underestimations of the OBB emissions (5 ~ 7 times) and the resultant $PM_{2.5}$ pollution (> 50 %) during EP2, the impacts of aerosol-radiation interactions become irrelevant here. More importantly, the associated process analysis has no benefit for addressing the main task of this study. Looking forward, it is necessary to advance the scientific understandings of the role of the constrained OBB-induced

aerosol-radiation interactions in disturbing the local weather systems as well as surface $PM_{2.5}$ pollution. Considering the distinct topic, we should combine sufficiently specific observations to give impetus to comprehensive process analysis, which will be the topic of a next separate study.

It is challenging to detect small fires that are far from fire hotspots. Thus, one should take more satellite-based burning products into consideration. Besides, geostationary satellites might be the prospective means to overcome the limitation of time spans

of polar-orbiting satellites. In addition, precise simulations of smoke plume and related chemical reactions are both challenging for regional and global models. The constraining method combining model results and available observations could be an effective way to reduce large uncertainties in the OBB emission inventories. Furthermore, uniform optimal factors for all species of OBB emissions are used in this study. It is necessary to conduct species-specific adjustments in order to accurately investigate OBB effects. Similarly, province-specific, even grid-specific adjustments should also be further explored.

Therefore, to improve the model capability in quantifying OBB and OCSB effects on regional air quality, more comprehensive experiments, field measurements and modelling efforts are called for.

*Data availability*. The MODIS data can be freely accessed at https://earthdata.nasa.gov/ (last access: 5 August 2019). GFASv1.0 data are available from http://apps.ecmwf.int/datasets/data/cams-gfas/ (last access: 5 August 2019). GFED4s data

can be downloaded from https://daac.ornl.gov/VEGETATION/guides/fire_emissions_v4.html (last access: 5 August 2019). FINNv1.5 data can be found at http://bai.acom.ucar.edu/Data/fire/ (last access: 5 August 2019).

*Supplement*. The supplement related to this article is available online.

*Author contributions*. SY and PL designed this study; SY, PL, KM, YW, LW, JS carried out analyses, interpreted data and wrote the manuscript. XC, ZL, YZ, ML, WL, YZ, DR contributed to the discussion. All authors have no competing interests.

*Competing interests*. The authors declare that they have no conflict of interest.

*Acknowledgements*. This study is supported by the Department of Science and Technology of China (No. 2016YFC0202702, 2018YFC0213506 and 2018YFC0213503), National Research Program for Key Issues in Air Pollution Control in China (No. DQGG0107) and National Natural Science Foundation of China (No. 21577126 and 41561144004). Pengfei Li is supported by Initiation Fund for Introducing Talents of Hebei Agricultural University (412201904). We acknowledge the great support of SCAS-CERN, Institute of Atmospheric Physics, Chinese Academy of Sciences (CAS) for providing the CARE-China data

for analysis.

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

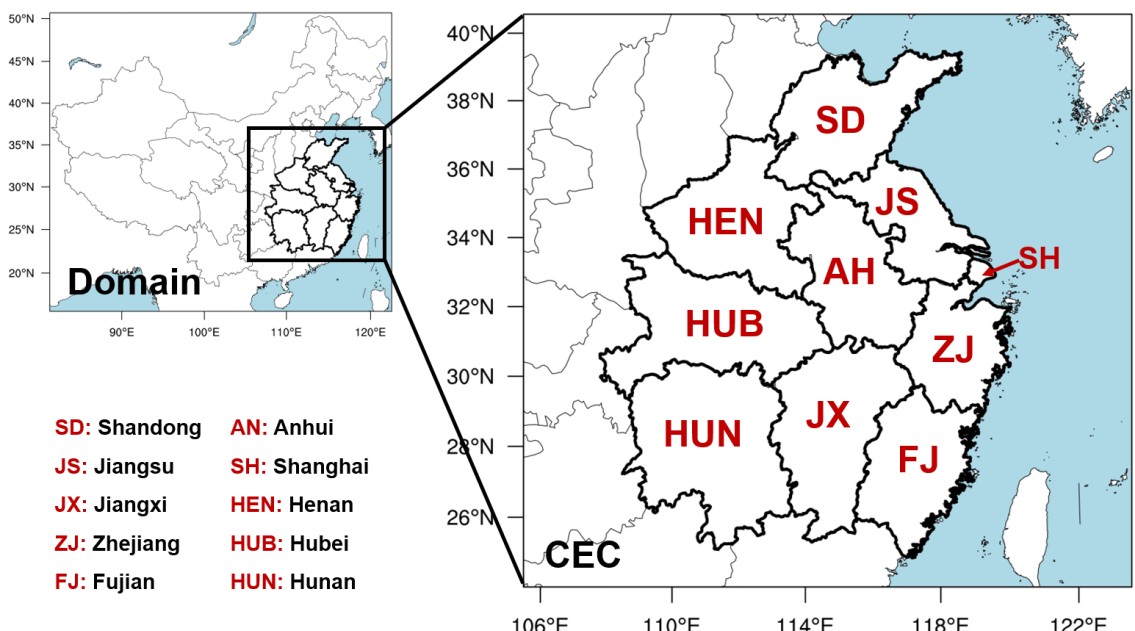

SD: Shandong    AN: Anhui

JS: Jiangsu    SH: Shanghai

JX: Jiangxi    HEN: Henan

ZJ: Zhejiang    HUB: Hubei

FJ: Fujian    HUN: Hunan

**Figure 1. Model domain and geographical areas of CEC. Black thick lines outline boundaries of 10 provinces belonging to CEC, including Anhui (AH), Hubei (HB), Henan (HEN), Hunan (HUN), Shandong (SD), Zhejiang (ZJ), Jiangsu (JS), Shanghai (SH), Fujian (FJ) and Jiangxi (JX).**

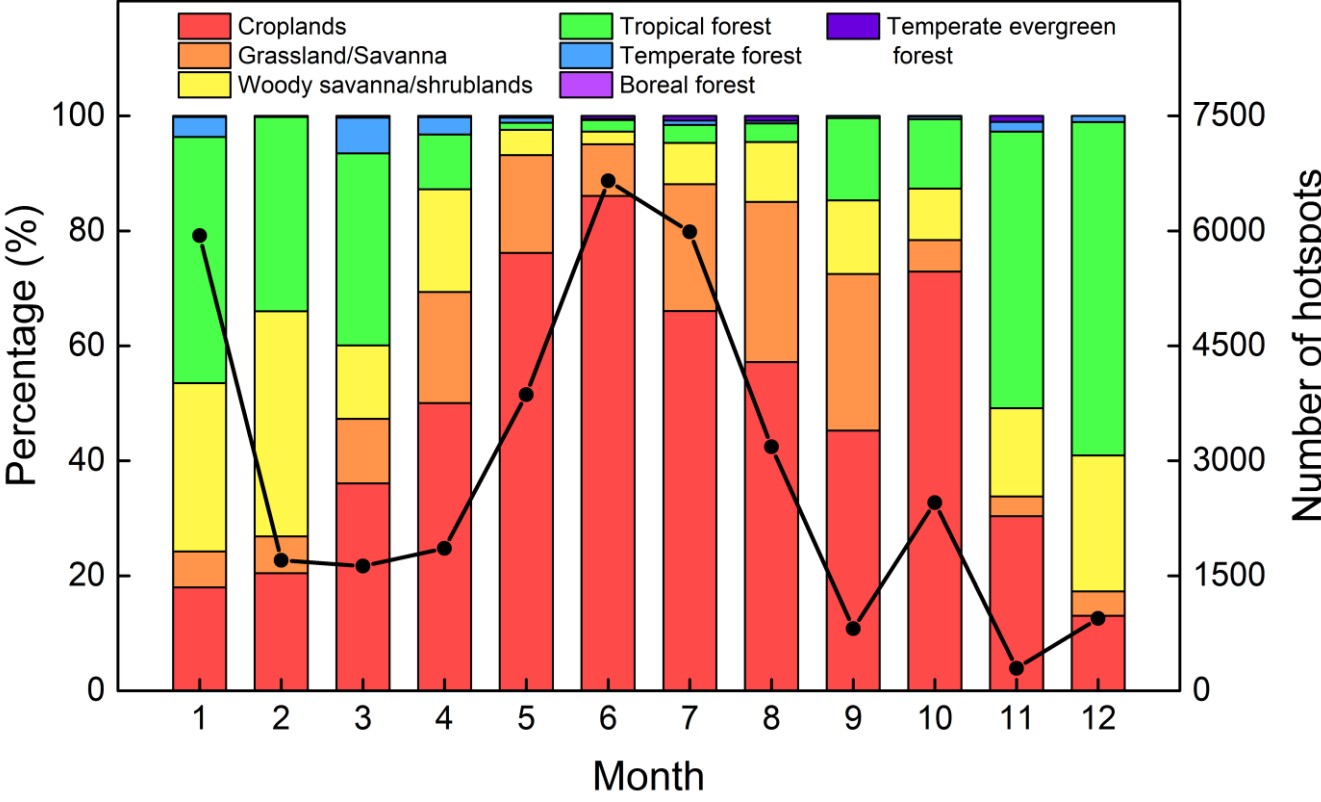

**Figure 2. Monthly variations in numbers of OBB fire hotspots in FINNv1.5 over CEC (right axis) in 2014 and relative contributions of seven types of OBB classified by corresponding land use types (left axis).**

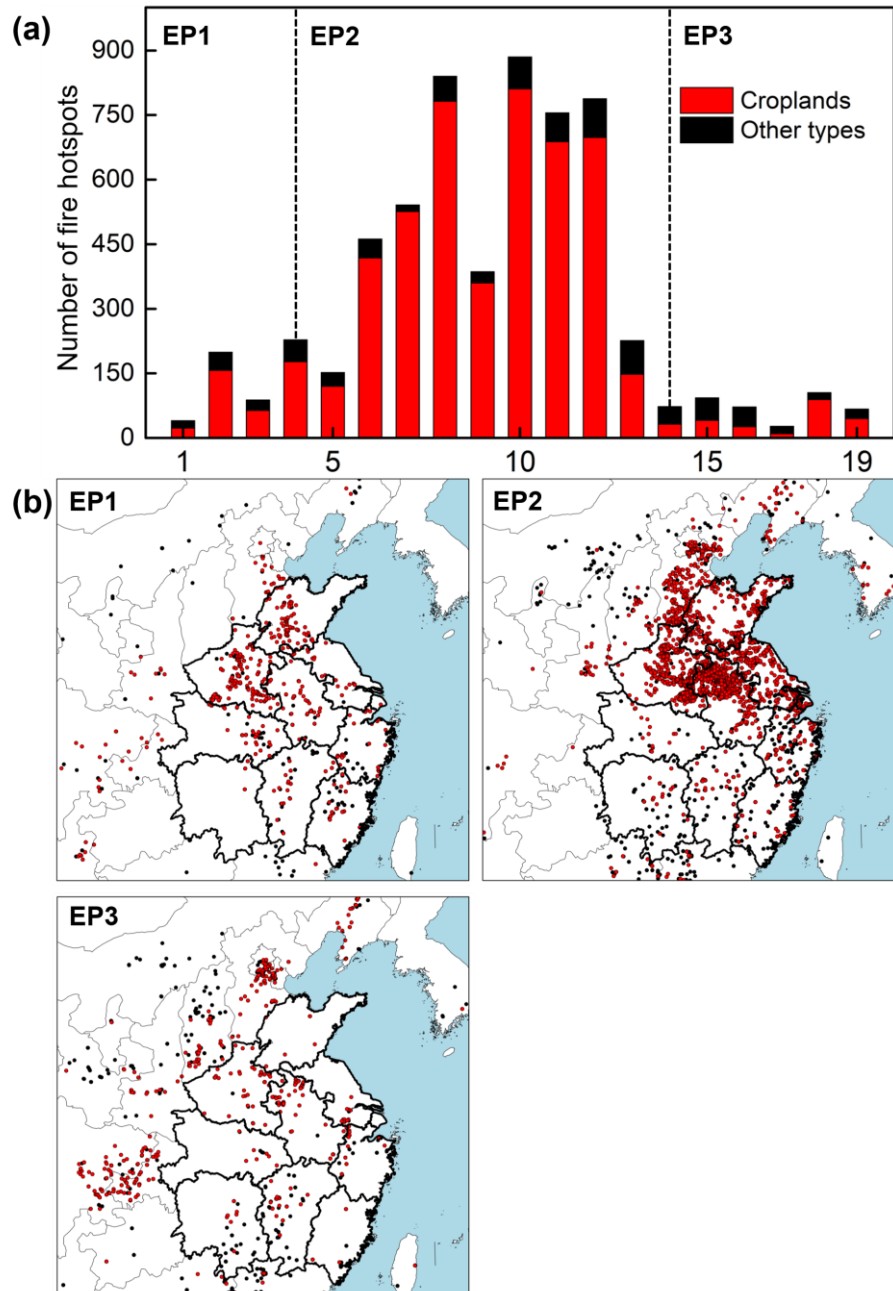

Figure 3. (a) Temporal and (b) spatial variations in numbers of fire hotspots for OCSB and other types of OBB during EP1, EP2 and EP3, which represent three successive episodes in turn, namely, from June 1 to 4, June 5 to 14, and June 15 to 19, respectively.

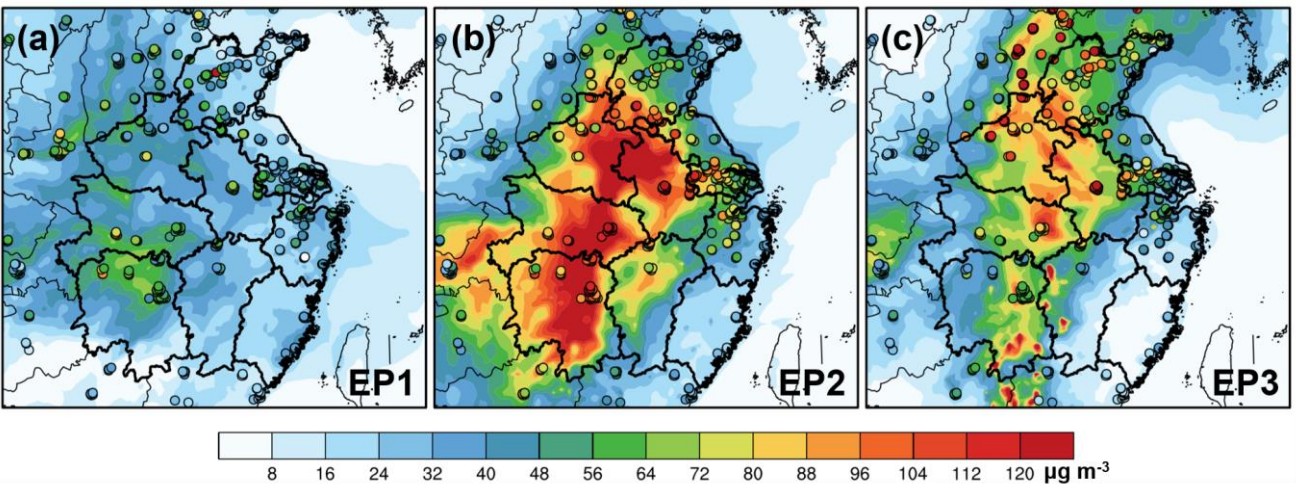

**Figure 4. Spatial distributions of simulated and observed episode-averaged PM₂.₅ concentrations over CEC during (a) EP1, (b) EP2, and (c) EP3. Colored circles denotes locations of ground measurement sites and corresponding values.**

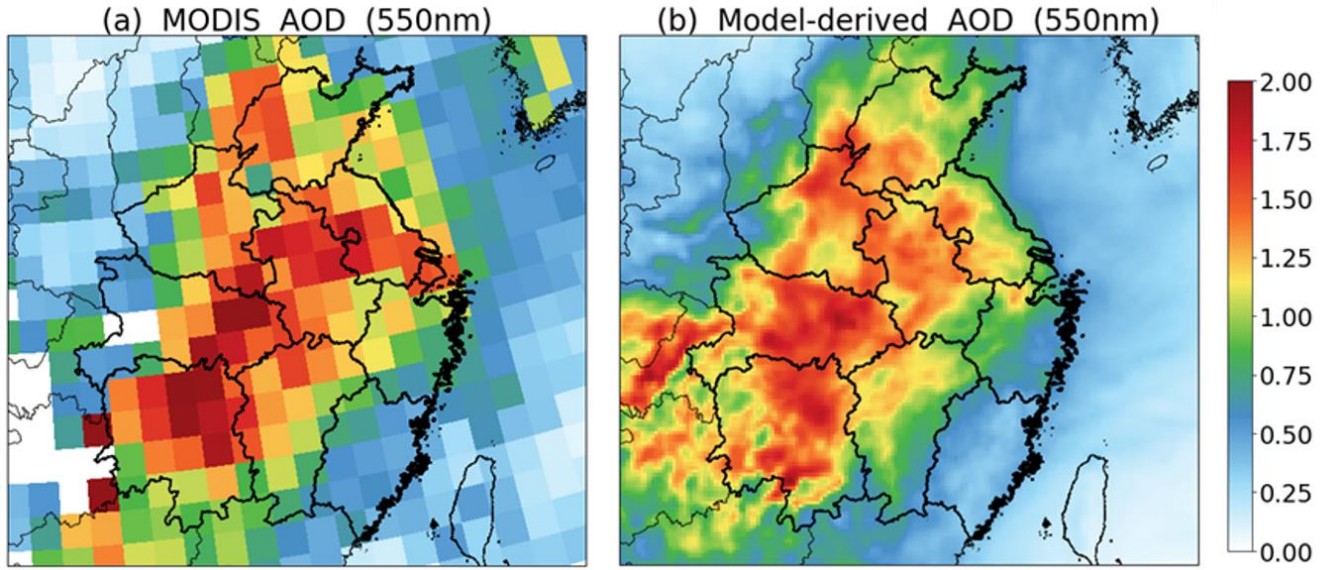

**Figure 5. Spatial distributions of (a) satellite-based and (b) model-derived AODs in the OPT case over CEC for EP2.**

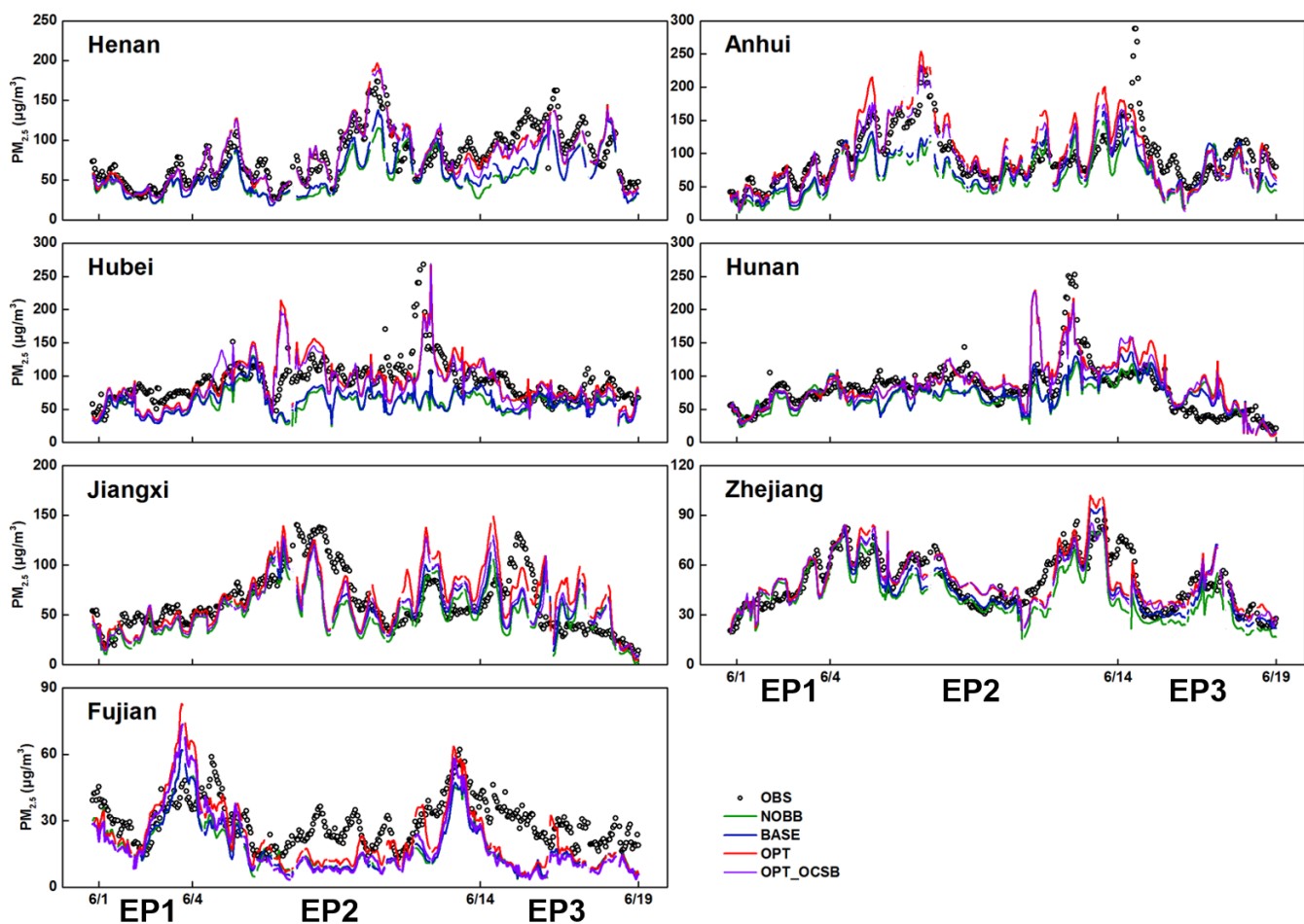

**Figure 6. Provincial average temporal variations in ground observed and simulated hourly PM$_{2.5}$ concentrations in Henan, Anhui, Hubei, Hunan, Jiangxi, Zhejiang and Fujian. The averaged concentrations for each province were calculated with values at the monitoring stations from both observations and different model simulations (OBS: observations; NOBB: the simulations with only anthropogenic and biogenic emissions; BASE: the simulations with not only anthropogenic and biogenic emissions but also original OBB emissions in FINNv1.5; OPT: the simulations with not only anthropogenic and biogenic emissions but also constrained OBB emissions; OPT_OCSB: the simulations with not only anthropogenic and biogenic emissions but also constrained OCSB emissions)**

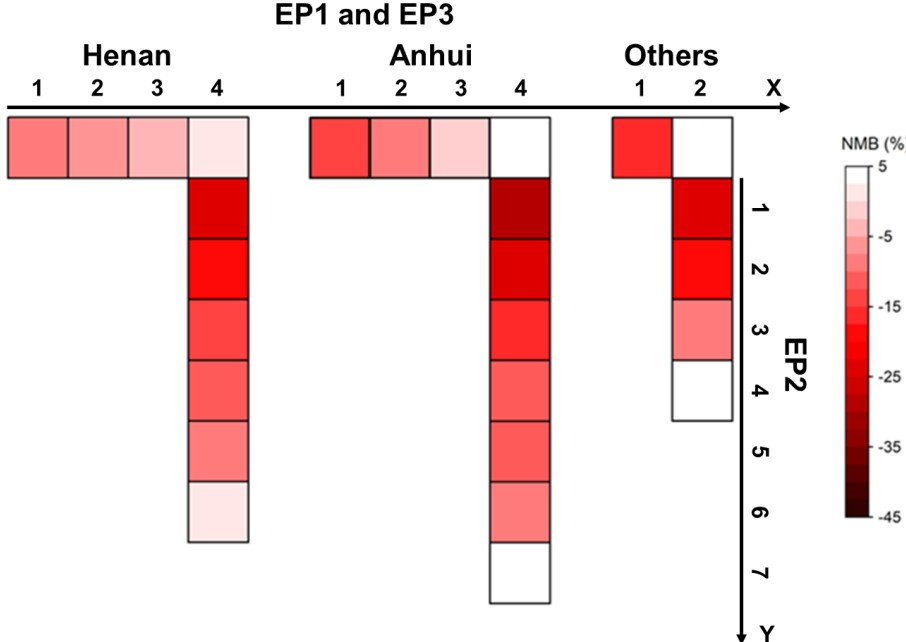

**Figure 7. Model responses to dynamic emission perturbations. Colored cells represent NMBs between observed and simulated PM$_{2.5}$ concentrations in different model sensitive tests for Henan, Anhui and other provinces over CEC. Coefficients in X axis refer to the adjustment coefficients for OBB emissions in EP1 and EP3, while coefficients in Y axis denotes the ones for EP2.**

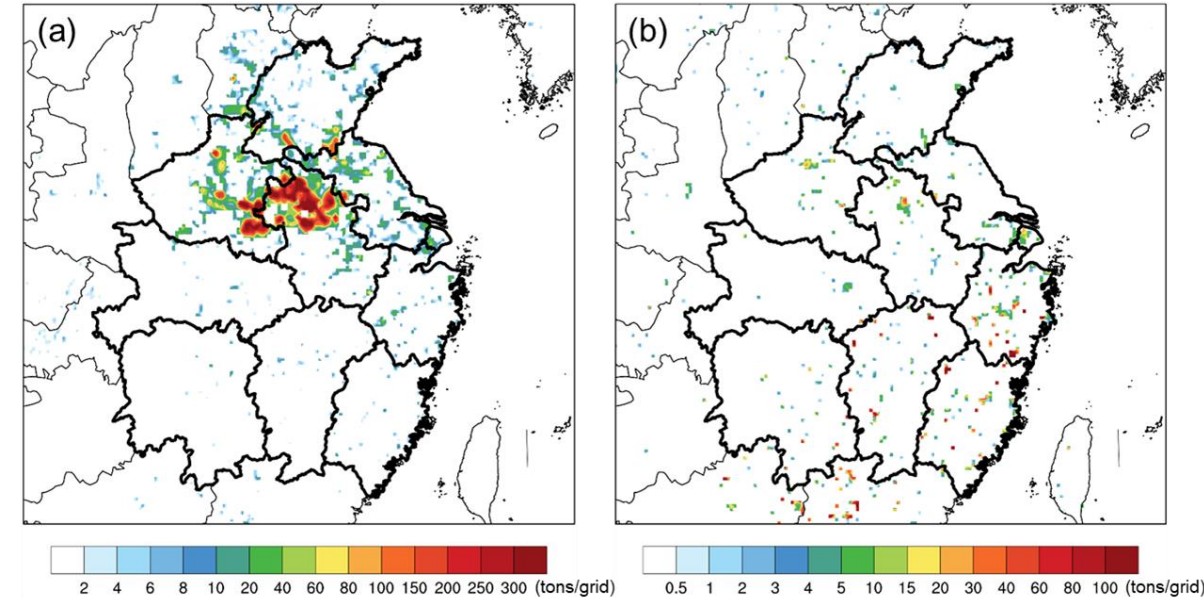

**Figure 8. Spatial distributions of constrained PM₂.₅ emissions from (a) OCSB and (b) other types of OBB for the study period.**

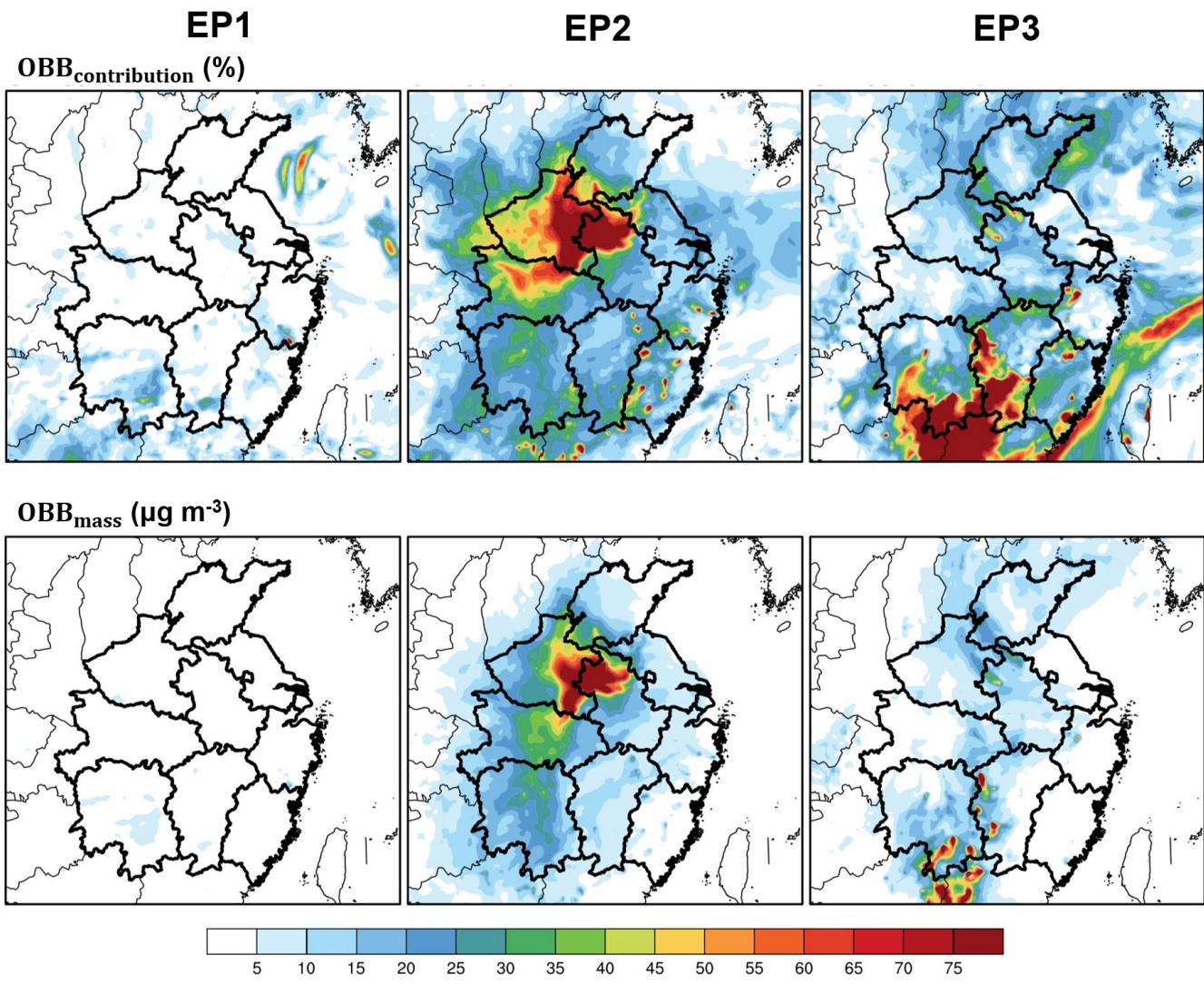

**Figure 9. Spatial distributions of OBB$_{mass}$ and OBB$_{contribution}$ during EP1, EP2 and EP3.**

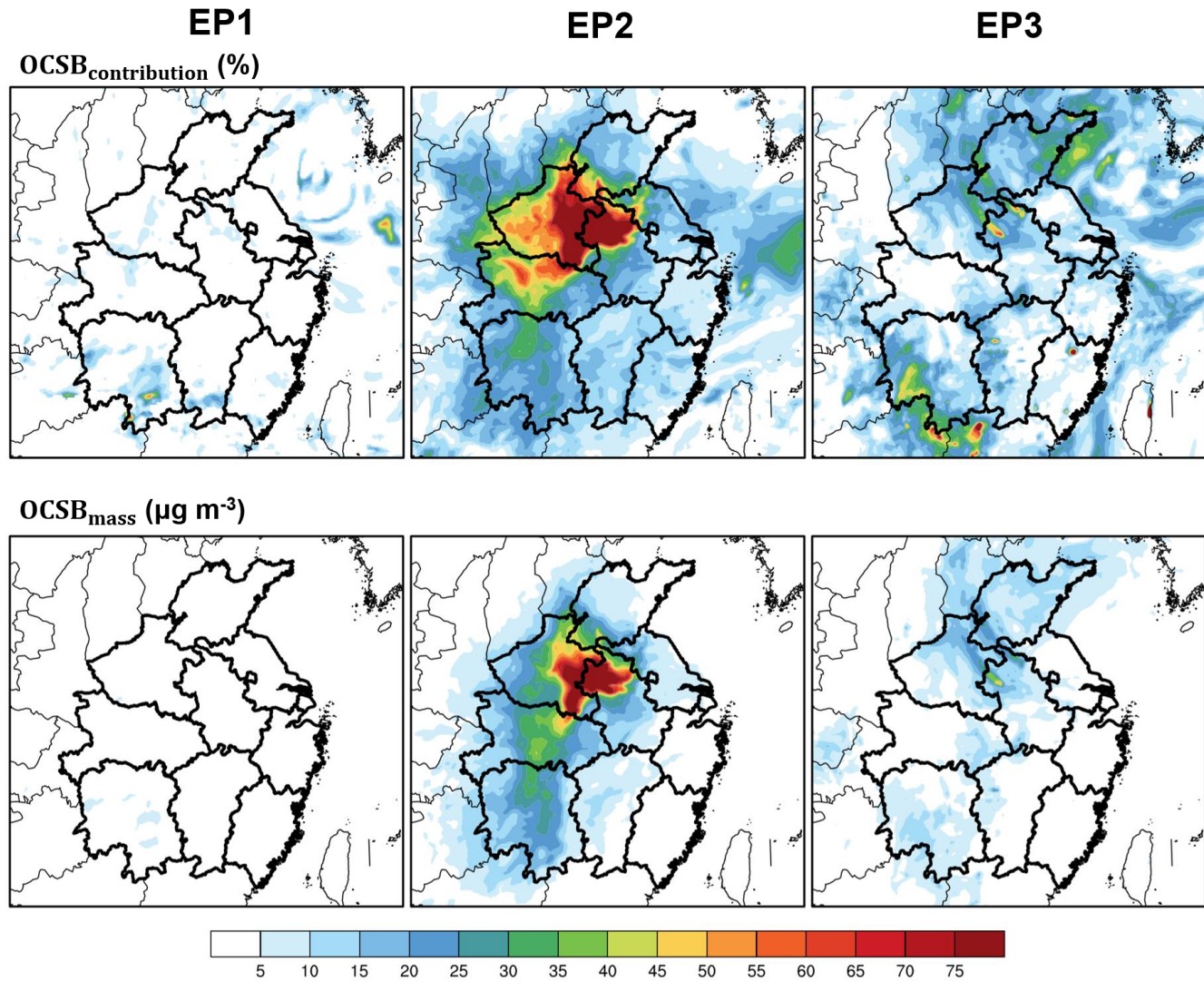

**Figure 10. The same as Fig. 9 but for OCSB$_{mass}$ and OCSB$_{contribution}$.**

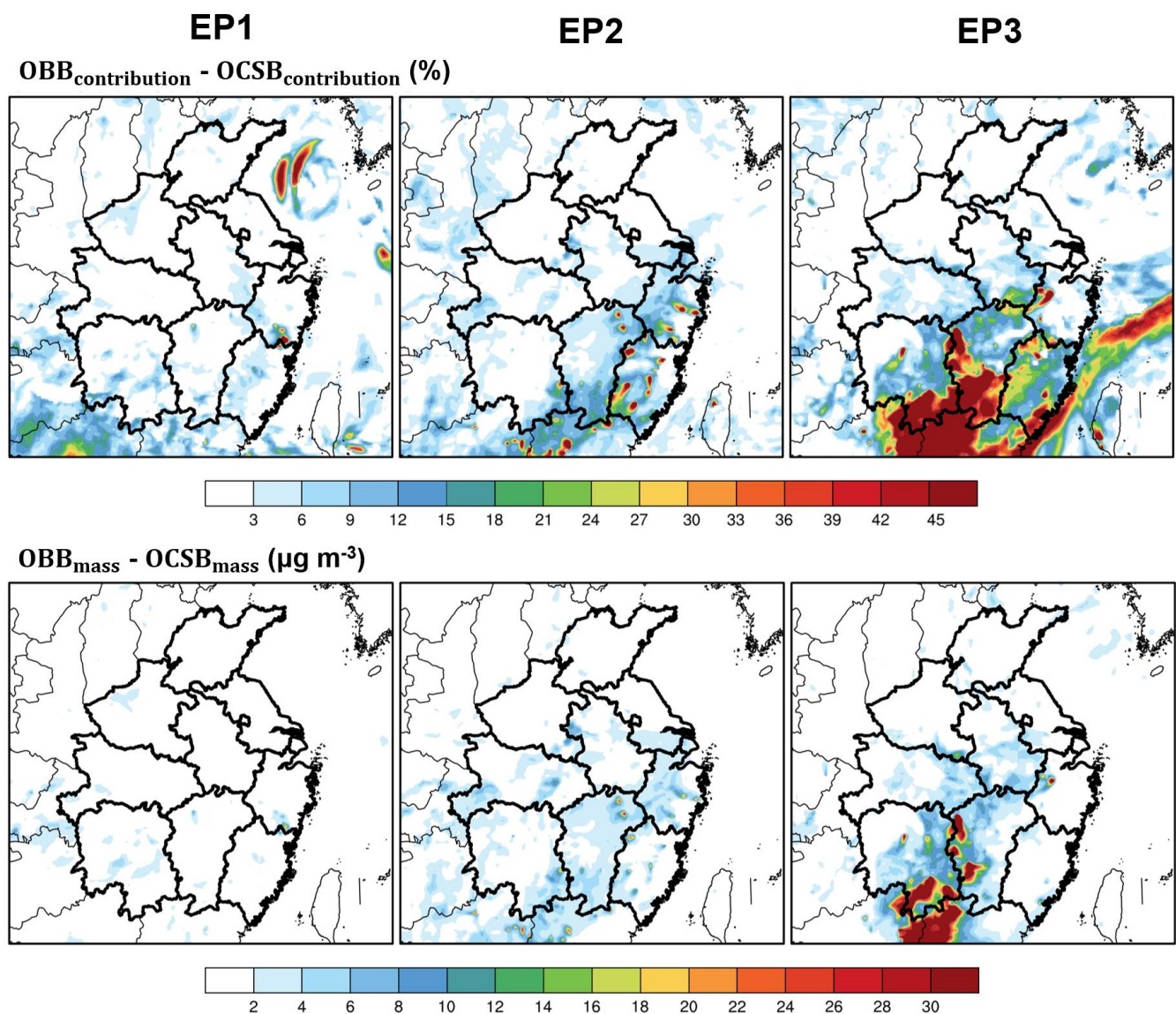

**Figure 11. The same as Fig. 10 but for the differences between OBB$_{mass}$ (OCSB$_{mass}$) and OBB$_{contribution}$ (OCSB$_{contribution}$).**

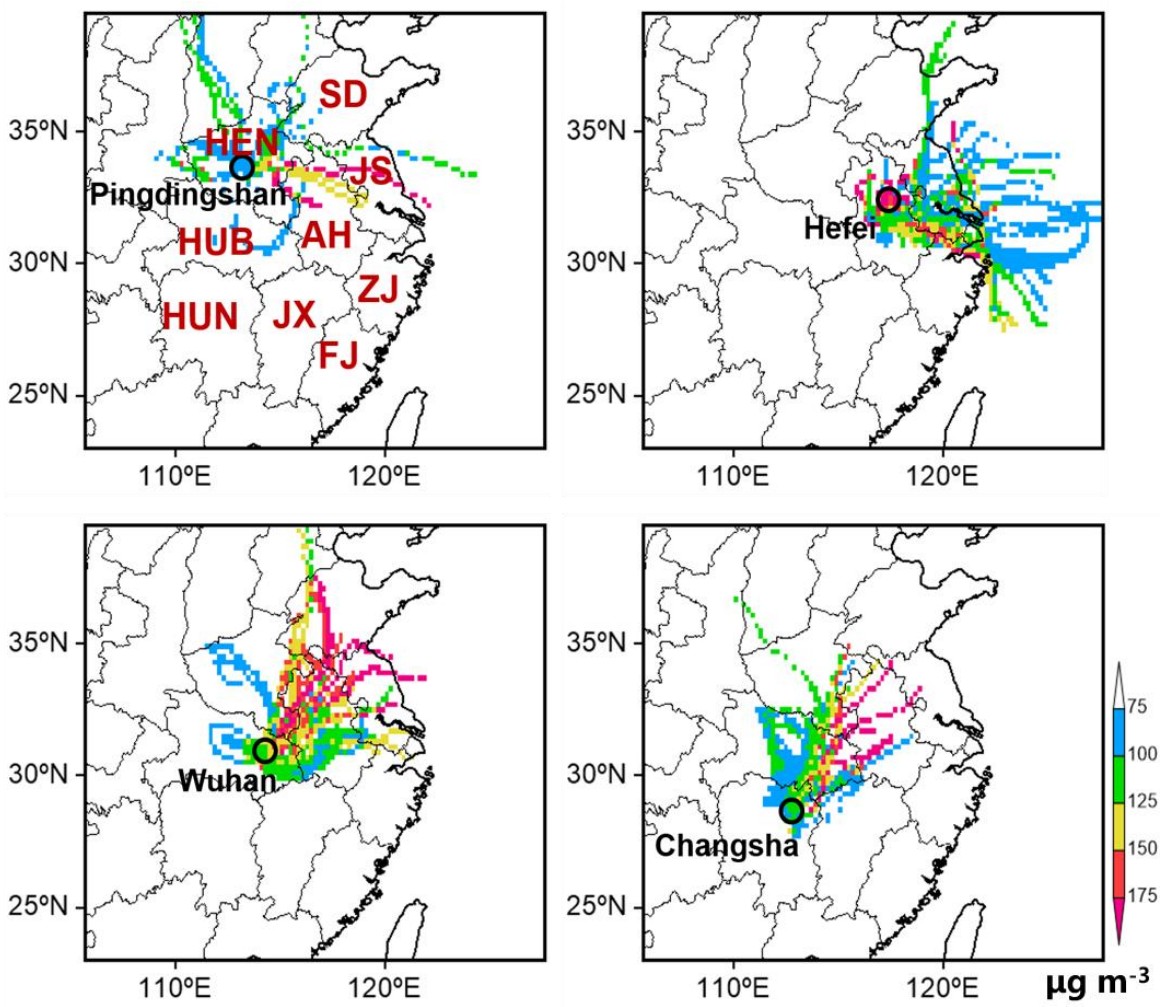

**Figure 12. CWT maps during EP2 at four representative cities (Pingdingshan, Hefei, Wuhan and Changsha) that represent Henan, Anhui, Hubei and Hunan, respectively.**

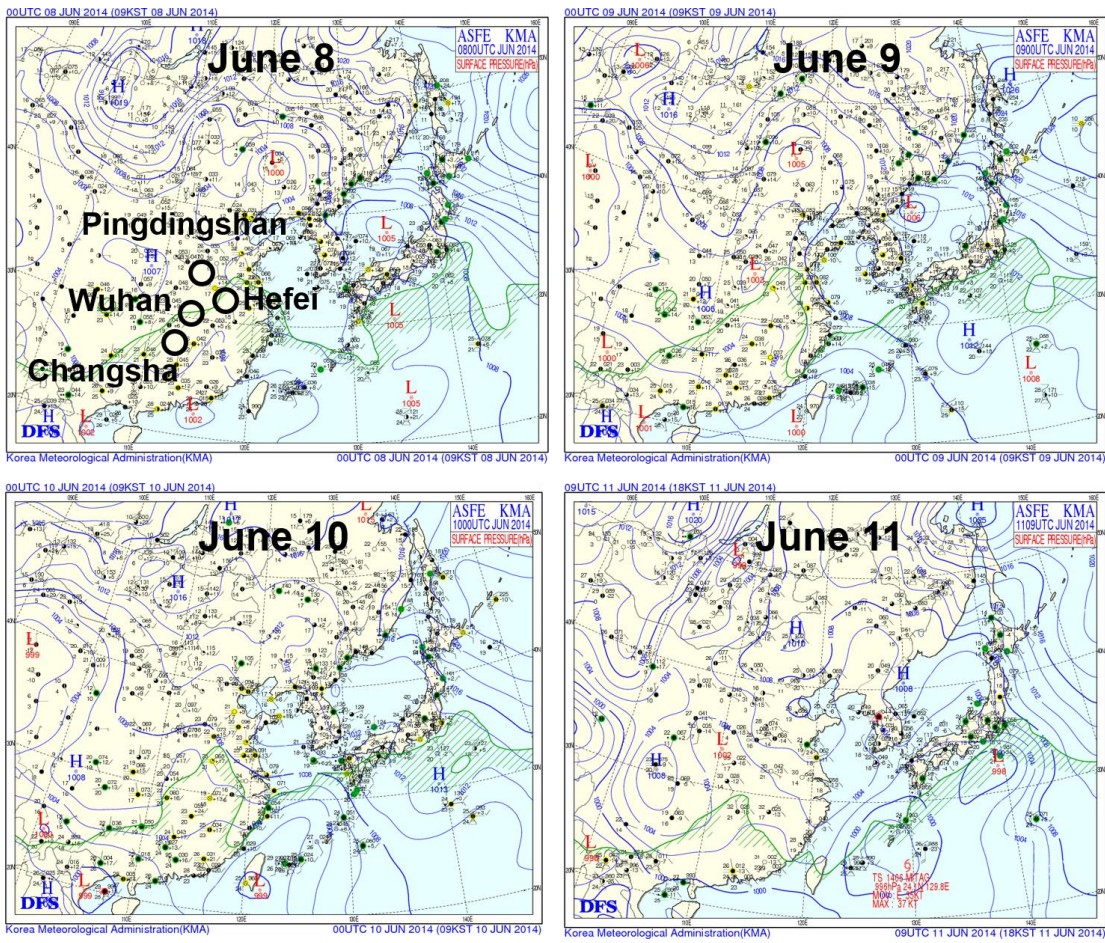

**Figure 13. Surface weather maps for June 8, June 9, June 10, and June 11 over CEC.**

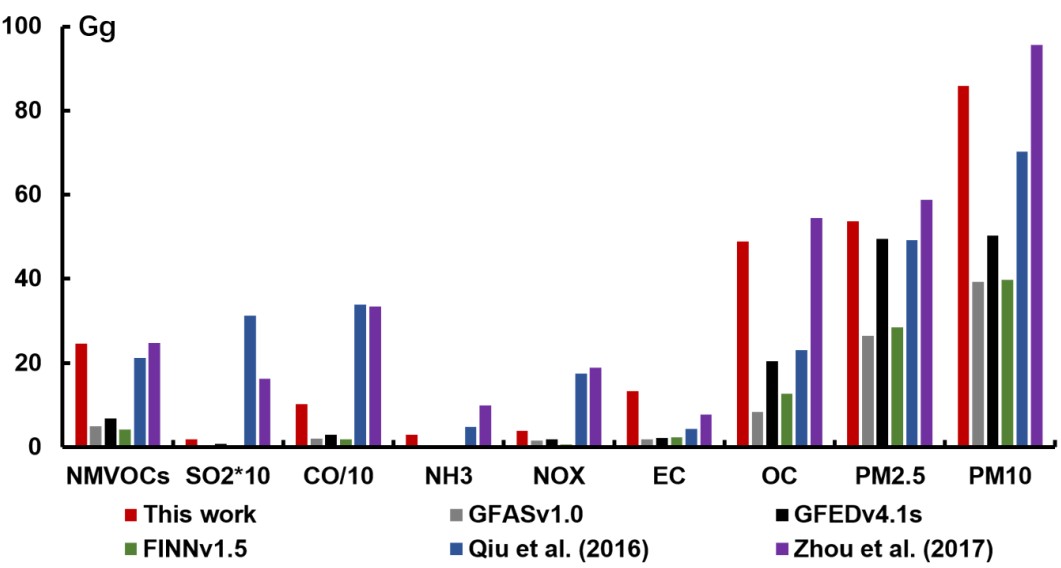

**Figure 14. Comparisons of OBB emissions between other studies and this study.**

**Table 1. The constrained OBB and OCSB emissions for EP1, EP2, and EP3 for each province over CEC (Units: million moles for NMVOCs, $SO_2$, CO, $NH_3$, and $NO_X$ and tons for EC, OC, primary $PM_{2.5}$ and $PM_{10}$).**

| Episode | Provinces | OBB | | | | | | | | | OCSB | | | | | | | | |
|---|---|---|---|---|---|---|---|---|---|---|---|---|---|---|---|---|---|---|---|
| | | NMVOCs | SO₂ | CO | NH₃ | NOx | EC | OC | PM₂.₅ | PM₁₀ | NMVOCs | SO2 | CO | NH₃ | NOx | EC | OC | PM₂.₅ | PM₁₀ |
| EP1 | Shanghai | 0 | 0 | 0 | 0 | 0 | 0 | 0 | 0 | 0 | 0 | 0 | 0 | 0 | 0 | 0 | 0 | 0 | 0 |
| | Jiangsu | 16 | 0 | 58 | 2 | 2 | 10 | 49 | 88 | 107 | 16 | 0 | 56 | 2 | 2 | 10 | 46 | 82 | 99 |
| | Zhejiang | 59 | 1 | 364 | 12 | 9 | 51 | 575 | 1107 | 1268 | 21 | 0 | 76 | 3 | 3 | 13 | 63 | 111 | 134 |
| | Anhui | 66 | 0 | 245 | 8 | 9 | 43 | 219 | 387 | 473 | 64 | 0 | 226 | 8 | 8 | 39 | 188 | 331 | 401 |
| | Fujian | 43 | 1 | 237 | 7 | 8 | 37 | 322 | 609 | 727 | 25 | 0 | 90 | 3 | 3 | 16 | 75 | 131 | 159 |
| | Jiangxi | 34 | 1 | 197 | 6 | 6 | 31 | 291 | 527 | 622 | 20 | 0 | 72 | 2 | 3 | 13 | 60 | 106 | 128 |
| | Shandong | 74 | 0 | 271 | 9 | 10 | 47 | 233 | 415 | 507 | 72 | 0 | 255 | 9 | 9 | 44 | 213 | 374 | 452 |
| | Henan | 318 | 2 | 1155 | 39 | 43 | 201 | 978 | 1735 | 2111 | 315 | 2 | 1116 | 38 | 41 | 194 | 929 | 1633 | 1976 |
| | Hubei | 40 | 0 | 158 | 5 | 6 | 28 | 155 | 266 | 326 | 39 | 0 | 137 | 5 | 5 | 24 | 114 | 201 | 243 |
| | Hunan | 4 | 0 | 18 | 1 | 1 | 3 | 20 | 34 | 41 | 4 | 0 | 14 | 0 | 1 | 3 | 12 | 21 | 25 |
| EP2 | Shanghai | 60 | 1 | 262 | 8 | 10 | 47 | 284 | 492 | 609 | 55 | 0 | 194 | 7 | 7 | 34 | 162 | 284 | 344 |
| | Jiangsu | 1104 | 7 | 4061 | 134 | 151 | 707 | 3549 | 6299 | 7681 | 1074 | 6 | 3809 | 130 | 141 | 663 | 3172 | 5575 | 6747 |
| | Zhejiang | 397 | 7 | 2197 | 69 | 67 | 345 | 3129 | 5768 | 6785 | 242 | 1 | 857 | 29 | 32 | 149 | 714 | 1255 | 1519 |
| | Anhui | 9485 | 54 | 33835 | 1147 | 1251 | 5890 | 28467 | 50126 | 60747 | 9430 | 53 | 33447 | 1140 | 1237 | 5824 | 27852 | 48953 | 59250 |
| | Fujian | 241 | 5 | 1554 | 48 | 43 | 232 | 2496 | 4691 | 5556 | 83 | 0 | 295 | 10 | 11 | 51 | 246 | 432 | 523 |
| | Jiangxi | 128 | 3 | 817 | 26 | 22 | 121 | 1301 | 2441 | 2835 | 50 | 0 | 179 | 6 | 7 | 31 | 149 | 262 | 317 |
| | Shandong | 1311 | 8 | 4752 | 159 | 177 | 828 | 4019 | 7124 | 8667 | 1296 | 7 | 4597 | 157 | 170 | 800 | 3828 | 6729 | 8144 |
| | Henan | 5000 | 29 | 17895 | 605 | 664 | 3116 | 15002 | 26467 | 32104 | 4975 | 28 | 17644 | 601 | 652 | 3072 | 14693 | 25824 | 31256 |
| | Hubei | 64 | 1 | 271 | 9 | 9 | 45 | 299 | 538 | 645 | 54 | 0 | 193 | 7 | 7 | 34 | 161 | 282 | 342 |
| | Hunan | 71 | 1 | 421 | 13 | 12 | 64 | 641 | 1197 | 1405 | 34 | 0 | 120 | 4 | 4 | 21 | 100 | 175 | 212 |
| EP3 | Shanghai | 0 | 0 | 0 | 0 | 0 | 0 | 0 | 0 | 0 | 0 | 0 | 0 | 0 | 0 | 0 | 0 | 0 | 0 |
| | Jiangsu | 15 | 0 | 58 | 2 | 2 | 10 | 58 | 103 | 125 | 14 | 0 | 48 | 2 | 2 | 8 | 40 | 70 | 85 |
| | Zhejiang | 24 | 0 | 129 | 4 | 4 | 19 | 186 | 355 | 411 | 12 | 0 | 43 | 1 | 2 | 7 | 36 | 63 | 76 |
| | Anhui | 152 | 1 | 551 | 18 | 21 | 96 | 469 | 833 | 1012 | 149 | 1 | 530 | 18 | 20 | 92 | 441 | 775 | 939 |
| | Fujian | 7 | 0 | 40 | 1 | 2 | 7 | 53 | 95 | 118 | 5 | 0 | 16 | 1 | 1 | 3 | 14 | 24 | 29 |
| | Jiangxi | 59 | 1 | 393 | 13 | 10 | 57 | 643 | 1205 | 1394 | 21 | 0 | 73 | 2 | 3 | 13 | 61 | 107 | 129 |
| | Shandong | 14 | 0 | 55 | 2 | 2 | 10 | 49 | 90 | 111 | 13 | 0 | 48 | 2 | 2 | 8 | 40 | 70 | 84 |
| | Henan | 61 | 0 | 223 | 7 | 8 | 39 | 189 | 336 | 410 | 60 | 0 | 214 | 7 | 8 | 37 | 178 | 313 | 379 |
| | Hubei | 7 | 0 | 28 | 1 | 1 | 5 | 29 | 52 | 65 | 6 | 0 | 22 | 1 | 1 | 4 | 18 | 32 | 39 |

| **Hunan** | 55 | 2 | 399 | 13 | 10 | 56 | 678 | 1299 | 1496 | | 9 | 0 | 33 | 1 | 1 | 6 | 27 | 48 | 58 |