# Peer review of "Relative effects of open biomass and crop straw burning on haze formation over central and eastern China: modelling study driven by constrained emissions"

_Atmospheric Chemistry and Physics, 2019_

## Referee Comment (RC1) · Anonymous Referee #1 · 7 Nov 2019

**General comments:**

Open biomass burning (OBB) is one of the major air pollution sources in many regions including central and eastern China (CEC). However, it's challenging to accurately estimate its emission amounts using either bottom-up or top-down methods. The bottom-up method suffers from large uncertainties in estimation components such as surface fuel loading, fuel consumption, and emission factors, while the top-down method has difficulties in detecting small fires like open crop straw burning (OCSB) studied in this work. Here the authors used a two-way coupled chemical transport

model (WRF-CMAQ) as well as ground and satellite observations to constrain a presumably biased satellite-based fire emission inventory (FINNv1.5) for the CEC region during a pollution episode in June 2014. They also evaluated regional air quality impacts of biomass burning based on the optimized OBB/OCSB emissions. The topic is within the scope of the journal, and the manuscript is well-organized and -written. However, some concerns regarding the generalization of its method and results exist. Therefore, I suggest its publication in the journal after addressing the following issues.

(1) The first and the biggest concern is about its scientific significance. This study mainly focused on a short time period less than a month. The major OBB/OCSB burning episode is about 10 days in EP2 from June 5 to 14, 2014. The authors spent their most efforts on scaling fire emissions in three time periods (EP1-3) and different CEC regions (Henan, Anhui, and other provinces over CEC) to reduce the normalized mean bias (NMB) as a metric of modeling performance. Since the studying time period is relatively short, this approach tends to fall into the "overfitting" problem as a common modeling error. It's questionable how robust are these scaling factors as shown in Fig. 7 as they are even distinct in different studying time periods (EP2 vs. EP1/3). It would be more interesting to extend the time scale and produce more generalizable optimization of biomass burning emissions in CEC with increased scientific merit.

(2) Regarding the research methodology, the authors essentially used a "trial and error" method to approach an optimal estimate of regional OBB emissions that led to a relatively good agreement between PM2.5 simulations and ground observations. Though it is straightforward by tweaking adjustment coefficients of total OBB emissions, this method has several limitations such as computationally expensive and cumbersome, indistinguishable bias sources, and possible over-adjustment. Given large uncertainties in many aspects of PM2.5 simulations, it is easy to ascribe the modeling discrepancy to wrong causes and correct the model to get a good-looking result for some wrong reason. It is suggested to do more comprehensive model evaluation in terms of aerosol speciation (to help identify OBB/non-OBB source contributions) and spatial distributions (both horizontal and vertical) before correcting

[Figure]

OBB primary emissions.

(3) A two-way coupled WRF-CMAQ model was used in this study, which considered aerosol-radiation interactions inherently. However, there is no discussion about this at all in the results and discussion section. The authors list the advantages of this fully coupled model by referring to a series of previous studies in the method section, then this key feature seems to be forgotten in the following sections. Readers would be interested in many questions related with aerosol feedbacks, such as how important those aerosol-radiation interactions are in the regional pollution episode, can they affect local weather systems and pollution severity significantly, etc.

Some technical corrections are listed below.

(1) Please add specific values for these parameters in Eq. (1) and Eq. (2).

(2) In Fig. 4, please indicate which modeling experiment is the PM2.5 simulation based on.

(3) In Fig. 12, please add unit in the label bar.

(4) Table 1 looks fuzzy due to the low resolution. Please improve the presentation quality.

---

## Referee Comment (RC2) · Anonymous Referee #2 · 11 Nov 2019

The paper by Mehmood et al. investigates the relative effects of open biomass burning (OBB) and open crop straw burning (OCSB) on haze formation, specifically surface PM2.5 mass concentrations, in central end eastern China. The authors used a fully coupled meteorological and chemical transport model (WRF-CMAQ), constrained by PM2.5 measurements made in a wide area, to derive the optional OBB emission rates based on the FINNv1.5 inventory. They show that the model simulation of PM2.5 improved significantly with the corrected FINNv1.5 inventory. The study is interesting and should be a welcome addition to the literature. The paper is well written in general and

can be accepted for publication before the following issues be addressed.

Specific comments:

- While OBB activities took place in rural areas, mass concentrations of surface PM2.5 and other chemical species were measured in the cities for this study. Is the grid resolution of the WRF-CMAQ model fine enough to capture the emissions and chemistry in the urban areas?

- The MODIS AOD dataset is used to show the haze distribution pattern in comparison with that of the model-simulated surface PM2.5 concentrations. How about the AOD distribution from the model? A comparison between the AODs from the model and MODIS would be interesting. The analysis of OMI AOD data might be skipped over due to so many default values.

Technical issues:

Abstract: It may be difficult for the readers who are not familiar with the Chinese geography to follow the descriptions using the province names.

---

## Author Comment (AC1) · 20 Jan 2020

**General comments:**

Open biomass burning (OBB) is one of the major air pollution sources in many regions including central and eastern China (CEC). However, it's challenging to accurately estimate its emission amounts using either bottom-up or top-down methods. The bottom-up method suffers from large uncertainties in estimation components such as surface fuel loading, fuel consumption, and emission factors, while the top-down method has difficulties in detecting small fires like open crop straw burning (OCSB) studied in this work. Here the authors used a two-way coupled chemical transport model (WRF-CMAQ) as well as ground and satellite observations to constrain a presumably biased satellite-based fire emission inventory (FINNv1.5) for the CEC region during a pollution episode in June 2014. They also evaluated regional air quality impacts of biomass burning based on the optimized OBB/OCSB emissions. The topic is within the scope of the journal, and the manuscript is well-organized and -written. However, some concerns regarding the generalization of its method and results exist. Therefore, I suggest its publication in the journal after addressing the following issues.

**Response:** We thank the reviewer #1 for the constructive comments and address them as below.

**Specific comments:**

1. The first and the biggest concern is about its scientific significance. This study mainly focused on a short time period less than a month. The major OBB/OCSB burning episode is about 10 days in EP2 from June 5 to 14, 2014. The authors spent their most efforts on scaling fire emissions in three time periods (EP1~3) and different CEC regions (Henan, Anhui, and other provinces over CEC) to reduce the normalized mean bias (NMB) as a metric of modelling performance. Since the studying time period is relatively short, this approach tends to fall into the "overfitting" problem as a common modelling error. It's questionable how robust are these scaling factors as shown in Fig. 7 as they are even distinct in different studying time periods (EP2 vs. EP1/3). It would be more interesting to extend the time scale and produce more generalizable optimization of biomass burning emissions in CEC with increased scientific merit.

**Response:** Yes, the scaling factors are specific to this case study. In this paper, we focus on a very representative period (i.e., EP2) when the abundant OCSB and other types of OBB outbreak concurrently, and the resultant $PM_{2.5}$ pollution spread

extensively. Further, we have achieved a valuable finding that is the degree of the uncertainties associated with these scaling factors for this period. That is to say, this paper has found a very important result in this regard.

35

2. Regarding the research methodology, the authors essentially used a "trial and error" method to approach an optimal estimate of regional OBB emissions that led to a relatively good agreement between $PM_{2.5}$ simulations and ground observations. Though it is straightforward by tweaking adjustment coefficients of total OBB emissions, this method has several limitations such as computationally expensive and cumbersome, indistinguishable bias sources, and possible over-adjustment. Given large

40 uncertainties in many aspects of $PM_{2.5}$ simulations, it is easy to ascribe the modelling discrepancy to wrong causes and correct the model to get a good-looking result for some wrong reason. It is suggested to do more comprehensive model evaluation in terms of aerosol speciation (to help identify OBB/non-OBB source contributions) and spatial distributions (both horizontal and vertical) before correcting OBB primary emissions.

**Response:** Given large uncertainties in previous OBB emissions (e.g., FINNv1.5), the "trial and error" method is relatively

45 necessary to optimize those estimates in comparison to the immediate predictions without any constraint. Using this method, we have quantified the critical uncertainties in previous OBB emission estimates to a large extent, which is one of the most important findings in this study. To address the reviewer's concerns about the model performance on aerosol speciaiton, we add the evaluations of the simulated $PM_{2.5}$ compositions (i.e., $K^+$, $SO_4^{2-}$, $NO_3^-$, OC, and $NH_4^+$) in the BASE and OPT cases in Sect. 3.4. The observed $PM_{2.5}$ composition was obtained from the CARE-China network (Xin et al., 2015; Liu et al., 2018),

50 which is introduced in Sect. 2.1. Owing to their key features, including the discontiguous samples as well as the 48h temporal resolution, they are not able to support the time series analysis but can be used for the period evaluations. Besides the evaluations at the province scale, the above results at the site level are well complementary for the horizontal evaluations. In addition, considering the lack of directly vertical measurements associated with this study, we supplement the comparisons of the simulated AODs in the OPT case against the observations. This is mainly due to the AODs, as the integration of multiple

55 column properties, generally serve as the vertical proxies to examine the model performance. Correspondingly detailed information has been given an account in the response for the specific comment (2) from Referee #2.

[revised manuscript text omitted]

**Added Table S5:**

Table S5. The comparisons of the simulated PM$_{2.5}$ composition (µg m$^{-3}$) in the BASE and OPT cases with the observations (OBS) as well as their corresponding NMB values (%).

| Composition | Time periods | | Changsha Values | Changsha NMB | Hefei Values | Hefei NMB | Yucheng Values | Yucheng NMB | Qianyanzhou Values | Qianyanzhou NMB | Wuxi Values | Wuxi NMB |
|---|---|---|---|---|---|---|---|---|---|---|---|---|
| K+ | From 10 a.m., June 2 to 10 a.m., June 4 | OBS | 0.59 | | 0.92 | | 0.90 | | 0.09 | | 0.62 | |
| | | BASE | 0.15 | -74.64 | 0.18 | -80.53 | 0.35 | -61.12 | 0.22 | 149.12 | 0.12 | -80.73 |
| | | OPT | 0.21 | -64.50 | 0.28 | -69.71 | 0.45 | -50.01 | 0.25 | 183.09 | 0.13 | -79.12 |
| | From 10 a.m., June 9 to 10 a.m., June 11 | OBS | 1.50 | | 1.16 | | 1.49 | | 0.56 | | 0.56 | |
| | | BASE | 0.49 | -67.34 | 0.55 | -52.49 | 0.78 | -47.67 | 0.41 | -26.80 | 0.38 | -32.23 |
| | | OPT | 0.79 | -47.34 | 0.89 | -23.12 | 0.87 | -41.63 | 0.45 | -19.65 | 0.34 | -39.37 |
| | From 10 a.m., June 16 to 10 a.m., June 18 | OBS | 0.51 | | 0.91 | | 0.51 | | 0.16 | | 0.25 | |
| | | BASE | 0.34 | -33.36 | 0.54 | -40.52 | 0.71 | 39.78 | 0.29 | 77.65 | 0.07 | -72.05 |
| | | OPT | 0.38 | -25.52 | 0.78 | -14.09 | 0.74 | 45.68 | 0.33 | 102.15 | 0.09 | -64.07 |
| SO$_4^{2-}$ | From 10 a.m., June 2 to 10 a.m., June 4 | OBS | 20.48 | | 10.95 | | 13.11 | | 12.56 | | 16.21 | |
| | | BASE | 11.40 | -44.34 | 5.90 | -46.12 | 8.70 | -33.64 | 1.98 | -84.24 | 9.64 | -40.54 |
| | | OPT | 24.12 | 17.77 | 13.54 | 23.64 | 15.34 | 17.01 | 3.45 | -72.53 | 9.58 | -40.91 |
| | From 10 a.m., June 9 to 10 a.m., June 11 | OBS | 33.83 | | 18.68 | | 30.74 | | 8.97 | | 13.23 | |
| | | BASE | 20.14 | -40.47 | 14.70 | -21.31 | 15.98 | -48.01 | 2.55 | -71.56 | 8.40 | -36.53 |
| | | OPT | 37.07 | 9.57 | 22.30 | 19.38 | 31.99 | 4.07 | 3.54 | -60.52 | 9.00 | -31.99 |
| | From 10 a.m., June 16 to 10 a.m., June 18 | OBS | 18.94 | | 27.05 | | 21.86 | | 20.95 | | 12.00 | |
| | | BASE | 13.75 | -27.39 | 16.70 | -38.26 | 8.71 | -60.15 | 4.78 | -77.19 | 6.85 | -42.90 |
| | | OPT | 23.64 | 24.83 | 28.90 | 6.85 | 18.05 | -17.41 | 5.61 | -73.23 | 7.40 | -38.31 |
| NO$_3^-$ | From 10 a.m., June 2 to 10 a.m., June 4 | OBS | 9.34 | | 8.48 | | 7.49 | | 1.93 | | 6.35 | |
| | | BASE | 10.97 | 17.49 | 5.39 | -36.44 | 3.61 | -51.81 | 1.03 | -46.56 | 1.78 | -71.96 |
| | | OPT | 14.98 | 60.44 | 9.81 | 15.67 | 6.45 | -13.90 | 2.62 | 35.94 | 1.95 | -69.28 |
| | From 10 a.m., June 9 to 10 a.m., June 11 | OBS | 11.57 | | 3.20 | | 12.72 | | 0.60 | | 1.82 | |
| | | BASE | 9.70 | -16.18 | 7.90 | 146.66 | 10.80 | -15.13 | 1.85 | 207.90 | 0.98 | -46.05 |
| | | OPT | 12.45 | 7.58 | 8.50 | 165.40 | 14.45 | 13.56 | 2.95 | 390.98 | 0.97 | -46.60 |
| | From 10 a.m., June 16 to 10 a.m., June 18 | OBS | 4.85 | | 28.33 | | 15.89 | | 2.22 | | 12.27 | |
| | | BASE | 7.40 | 52.45 | 20.00 | -29.39 | 20.38 | 28.23 | 1.88 | -15.36 | 14.95 | 21.82 |
| | | OPT | 8.78 | 80.88 | 29.80 | 5.21 | 21.09 | 32.69 | 2.23 | 0.40 | 15.78 | 28.59 |
| NH$_4^+$ | From 10 a.m., June 2 to 10 a.m., June 4 | OBS | 10.26 | | 9.03 | | 8.45 | | 5.46 | | 9.17 | |
| | | BASE | 8.45 | -17.63 | 7.54 | -16.53 | 3.78 | -55.27 | 1.03 | -81.14 | 13.50 | 47.27 |
| | | OPT | 11.54 | 12.50 | 10.92 | 20.89 | 7.30 | -13.61 | 1.75 | -67.96 | 13.90 | 51.64 |
| | From 10 a.m., June 9 to 10 a.m., June 11 | OBS | 17.68 | | 8.54 | | 15.26 | | 3.71 | | 4.76 | |
| | | BASE | 11.06 | -37.45 | 4.98 | -41.72 | 13.45 | -11.88 | 1.35 | -63.59 | 2.01 | -57.77 |
| | | OPT | 17.37 | -1.76 | 10.39 | 21.60 | 18.74 | 22.78 | 1.75 | -52.80 | 3.32 | -30.25 |
| | From 10 a.m., June 16 to 10 a.m., June 18 | OBS | 8.62 | | 21.87 | | 14.95 | | 10.14 | | 9.74 | |
| | | BASE | 3.84 | -55.47 | 20.06 | -8.29 | 14.20 | -4.99 | 5.97 | -41.11 | 8.70 | -10.65 |
| | | OPT | 10.43 | 20.94 | 28.78 | 31.57 | 17.64 | 18.03 | 6.78 | -33.12 | 8.40 | -13.73 |
| OC | From 10 a.m., June 9 to 10 a.m., June 11 | OBS | 19.80 | | 19.54 | | 18.95 | | 15.70 | | 29.15 | |
| | | BASE | 10.15 | -48.74 | 10.30 | -47.30 | 10.65 | -43.81 | 6.12 | -61.02 | 15.07 | -48.30 |
| | | OPT | 18.50 | -6.57 | 15.50 | -20.70 | 13.50 | -28.77 | 8.12 | -48.28 | 15.95 | -45.28 |

3. A two-way coupled WRF-CMAQ model was used in this study, which considered aerosol-radiation interactions inherently. However, there is no discussion about this at all in the results and discussion section. The authors list the advantages of this fully coupled model by referring to a series of previous studies in the method section, then this key feature seems to be forgotten in the following sections. Readers would be interested in many questions related with aerosol feedbacks, such as how important those aerosol-radiation interactions are in the regional pollution episode, can they affect local weather systems and pollution severity significantly, etc.

**Response:** We supplement the additional discussions in Sect. 4 to explain why we have not explored the effects of OBB-induced aerosol-radiation interactions during EP2. This is mainly because that the effects of the aerosol-radiation interactions are projected to be particularly small compared to the large uncertainties in the OBB emissions.

Specifically, the two-way coupled WRF-CMAQ utilized in this study is a standard approach. A lot of previous studies have applied the same or similar model to explore aerosol-radiation interactions. Generally, this process could enhance surface PM$_{2.5}$ concentrations by 8 ~ 12 % over CEC (Wang et al., 2014, 2015; Zhang et al., 2015, 2018; Jung et al., 2019). Compared

115  to the significant underestimations of the OBB emissions (5 ~ 7 times) and the resultant PM$_{2.5}$ pollution (> 50 %) during EP2, the impacts of aerosol-radiation interactions become irrelevant here. More importantly, the associated process analysis has no benefit for addressing the main task of this study.

**Added/rewritten part in Sect. 4:** To optimize the OBB emissions, previous attempts always focused on the primary emissions but neglected aerosol-radiation interactions (Uranishi et al., 2019; Yang and Zhao, 2019). Theoretically, with the aid of the
120  systematic considerations of aerosol-radiation interactions in the two-way coupled WRF-CMAQ model, this study was projected to achieve more reliable results. A lot of previous studies have applied the same or similar model to explore the impacts of aerosol-radiation interactions. However, it has been found that this process generally enhanced surface PM$_{2.5}$ concentrations by 8 ~ 12 % over CEC (Wang et al., 2014, 2015; Zhang et al., 2015, 2018; Jung et al., 2019). Compared to the significant underestimations of the OBB emissions (5 ~ 7 times) and the resultant PM$_{2.5}$ pollution (> 50 %) during EP2, the
125  impacts of aerosol-radiation interactions become irrelevant here. More importantly, the associated process analysis has no benefit for addressing the main task of this study. Looking forward, it is necessary to advance the scientific understandings of the role of the constrained OBB-induced aerosol-radiation interactions in disturbing the local weather systems as well as surface PM$_{2.5}$ pollution. Considering the distinct topic, we should combine sufficiently specific observations to give impetus to comprehensive process analysis, which will be the topic of a next separate study.

130

**Technical corrections:**

1. Please add specific values for these parameters in Eq. (1) and Eq. (2).

**Response:** We supplement specific values (i.e., Table S1 and Table S2) for B$_{size}$, B$_{hour}$, P$_{topmax}$, and P$_{bottommax}$ based on the look-up tables.

135  **Added/rewritten part in Sect. 2.2:** In this study, we determined the heights of the hourly top ($P_{top}$) and bottom ($P_{bottom}$) of the OBB plume using a quick plume rise model (Eq. (4) and Eq. (5)). This process was calculated based on the buoyant efficiency ($B$) available from the corresponding hourly and size class tables (Tables S1 and S2) (Tai et al., 2008; Fu et al., 2012a).

140

**Added Table S1:**

Table S1. The fire-related parameters (i.e., B$_{size}$, P$_{topmax}$, and P$_{bottommax}$) as a function of the fire size classes.

| Fire Class | 1 | 2 | 3 | 4 | 5 |
|---|---|---|---|---|---|
| Size | 0~10 | 10~100 | 100~1000 | 1000~5000 | >5000 |
| B$_{size}$ | 0.4 | 0.6 | 0.75 | 0.85 | 0.9 |
| P$_{topmax}$ | 160 | 2400 | 6400 | 7200 | 8000 |

| $P_{bottommax}$ | 0 | 900 | 2200 | 3000 | 300 |
|---|---|---|---|---|---|

145

**Added Table S2:**

Table S2. The activity related parameter (i.e., $B_{hour}$) as a function of hour of day.

| Hour | $B_{hour}$ | Hour | $B_{hour}$ |
|---|---|---|---|
| 0 | 0.03 | 12 | 0.7 |
| 1 | 0.03 | 13 | 0.8 |
| 2 | 0.03 | 14 | 0.9 |
| 3 | 0.03 | 15 | 0.95 |
| 4 | 0.03 | 16 | 0.99 |
| 5 | 0.03 | 17 | 0.8 |
| 6 | 0.03 | 18 | 0.7 |
| 7 | 0.03 | 19 | 0.4 |
| 8 | 0.06 | 20 | 0.06 |
| 9 | 0.1 | 21 | 0.03 |
| 10 | 0.2 | 22 | 0.03 |
| 11 | 0.4 | 23 | 0.03 |

2. In Fig. 4, please indicate which modelling experiment is the $PM_{2.5}$ simulation based on.

150 **Response:** The simulated $PM_{2.5}$ concentrations in Fig. 4 were extracted from the OPT case. We add the corresponding statement in Fig. 4.

**The rewritten caption in Fig. 4:** Spatial distributions of observed and simulated episode-averaged $PM_{2.5}$ concentrations from the OPT case over CEC during (a) EP1, (b) EP2, and (c) EP3. Colored circles denotes locations of ground measurement sites and corresponding values.

155

3. In Fig. 12, please add unit in the label bar.

**Response:** We add the units ($\mu g\ m^{-3}$) in the label bar of Fig. 12.

**The updated Fig. 12:**

[Figure]

Figure 12. CWT maps during EP2 at four representative cities (Pingdingshan, Hefei, Wuhan and Changsha) that represent Henan, Anhui, Hubei and Hunan, respectively.

4. Table 1 looks fuzzy due to the low resolution. Please improve the presentation quality.

**Response: We improve the presentation quality of Table 1.**

---

## Author Comment (AC2) · 20 Jan 2020

**General comments:**

The paper by Mehmood et al. investigates the relative effects of open biomass burning (OBB) and open crop straw burning (OCSB) on haze formation, specifically surface $PM_{2.5}$ mass concentrations, in central and eastern China. The authors used a fully coupled meteorological and chemical transport model (WRF-CMAQ), constrained by $PM_{2.5}$ measurements made in a wide area, to derive the optional OBB emission rates based on the FINNv1.5 inventory. They show that the model simulation of $PM_{2.5}$ improved significantly with the corrected FINNv1.5 inventory. The study is interesting and should be a welcome addition to the literature. The paper is well written in general and can be accepted for publication before the following issues be addressed.

**Response: We thank the reviewer #2 for the constructive comments and address them as below.**

**Specific comments:**

1. While OBB activities took place in rural areas, mass concentrations of surface $PM_{2.5}$ and other chemical species were measured in the cities for this study. Is the grid resolution of the WRF-CMAQ model fine enough to capture the emissions and chemistry in the urban areas?

**Response:** For both urban and rural areas, we adopted 12km as the horizontal grid resolution to resolve all relevant emission and chemical processes in the WRF-CMAQ model. To some extent, this resolution (i.e., 12km) is the typical setup, since previous studies have accumulated a wealth of similar experiences (Wu et al., 2018; Xing et al., 2018; Yu et al., 2018; Chen et al., 2019; Qiao et al., 2019). They have demonstrated that such configuration, together with reasonable settings for other numerical processes, could enable the model to derive reliable simulations for urban haze. Thus, this indirectly indicates that the horizontal grid resolution of 12km in the WRF-CMAQ model is virtually applicable to capturing the emissions and chemistry in the urban areas. On the other hand, several studies have directly explored the effects of horizontal grid resolution on urban haze using the WRF-CMAQ model (Gan et al., 2016). Also, the horizontal grid resolution of 12km has been found to be fine enough for characterizing local gradients for limited regions (e.g., urban areas). Yet, certain biases and uncertainties

still exist because it is the challenge for this resolution to resolve the numerical processes in urban microenvironment, such as the turbulent diffusion with chemical reactions under the convective boundary layer (Han et al., 2019).

35  2. The MODIS AOD dataset is used to show the haze distribution pattern in comparison with that of the model-simulated surface PM$_{2.5}$ concentrations. How about the AOD distribution from the model? A comparison between the AODs from the model and MODIS would be interesting. The analysis of OMI AOD data might be skipped over due to so many default values.

**Response:** We add the comparisons of simulated AOD for EP2 with the corresponding MODIS AOD datasets in Sec.3.4. Their respective spatial distributions are shown in the updated Figure 5. We find that the model could reproduce the

40  approximately spatial patterns of the observed AODs, in particular, the relatively high values spreading over Henan, Anhui, Hubei, and Hunan. In advance, we need to identify the calculation processes of the WRF-CMAQ-derived AODs, which would be supplemented in Sec.2.1.

Due to the excessive lack of the OMI AOD data, as the reviewer has pointed out, we rewrite all relevant descriptions and presentations that originally existed in Sec.2.4, Sect.3.2, Figure 5, and the statement of "Data availability". As abovementioned,

45  we further replenish the model-derived AODs to enhance observational evidence, as well as to validate the model performance in Sec.3.4.

**Added/rewritten part in Sect. 2.1:** To comprehensively validate the model performance, we would evaluate the spatial distributions of model-derived AODs, besides primary chemical and meteorological factors. Theoretically, not only particles but also gases have the ability to attenuate the intensity of light. AODs, generally severing as the feature of extinctions, should

50  be the combined function of their scattering and absorption. However, owing to the insignificant magnitude of gases, we focused only on particles to estimate the model-derived AODs as the following equations (Malm et al., 1994; Binkowski and Roselle, 2003; Song et al., 2008; Park et al., 2011; Jeon et al., 2016):

$$AOD_{MODEL} = \sum_{i=1}^{N} (\sigma_{sp} + \sigma_{ap}) \Delta Z_i \ (1)$$

$$\sigma_{sp} = 0.003 f(RH)(NH_4^+ + +SO_4^{2-} + NO_3^-) + 0.004 OM + 0.001 FS + 0.0006 CM \ (2)$$

55  $$\sigma_{ap} = 0.01 LAC \, , \ (3)$$

where i denoted to the vertical layer number and $Z_i$ referred to the corresponding layer thickness. The OM, FS, CM, and LAC were the mass concentrations of organic species, fine soil, coarse particles, and black carbon, respectively and uniformly configured with the units of mg/m$^3$. Their respective scattering and absorbing coefficients (i.e., 0.003, 0.004, 0.001, 0.0006, and 0.001) were recorded in m$^2$/mg. The f(RH) represented the aerosol growth factor that was estimated based on the relative

60  humidity. All relevant parameters were extracted from the model results.

**Added/rewritten part in Sect. 2.4:** Daily mean values of AOD at 550 nm retrieved from the satellite platform were examined during the target period to highlight significant spatial and temporal variabilities of regional haze in CEC. Here the episode-averaged AOD product from MODIS (MOD08_D3) at 550 nm was utilized (https://giovanni.sci.gsfc.nasa.gov /giovanni/, last access: 5 August 2019).

65 **Added/rewritten part in Sect. 3.2:** Figure 5 shows spatial distributions of episode-averaged AOD observed by MODIS (MOD08_D3) at 550 nm during EP2. It is in good agreement with spatial distributions of surface average PM$_{2.5}$ concentrations. For instance, much higher AOD values were mostly detected in Henan, Anhui, Hubei, and Hunan, associated with relatively high surface observed PM$_{2.5}$ concentrations and substantial OCSB emissions, as shown in Figs. 3 and 4. In addition, the satellite-based product detected that spatial distributions of high AOD values covered wider areas than the surface

70 measurements, such as in Jiangxi, Zhejiang and Fujian. This was possibly due to the fact that PM suspended in the upper troposphere was more easily transported than that on the ground. This phenomenon further illustrates that OBB dominated by OCSB is not only a significant local source but also an important regional source.

**Added/rewritten part in Sect. 3.4:** Besides, compared with the satellite retrievals, the model-derived AODs in the OPT case during EP2 presented the extremely similar spatial patterns over CEC (Fig. 5). Especially, they could reproduce the relatively

75 high measurements over Henan, Anhui, Hubei, and Hunan. Nevertheless, we recognized the general underestimations of model-derived AODs, in particular over the areas with the extremely PM$_{2.5}$ concentrations, which might be duo to the uncertainties in the numerical predictions of the plume rise of OBB (Tai et al., 2008; Fu et al., 2012a). Another explanation may be contamination of the observed AODs due to opaque clouds as described by several studies (Huang et al., 2012; Aouizerats et al., 2015). These results establish reliable model performance.

80 **The updated Fig.5:**

[Figure]

**Figure 5. Spatial distributions of (a) satellite-based and (b) model-derived AODs in the OPT case over CEC for EP2.**

**Added/rewritten part in "Data availability":** The MODIS data can be freely accessed at https://earthdata.nasa.gov/ (last

85 access: 5 August 2019). GFASv1.0 data are available from http://apps.ecmwf.int/datasets/data/cams-gfas/ (last access: 5

August 2019). GFED4s data can be downloaded from https://daac.ornl.gov/VEGETATION/guides/fire_emissions_v4.html (last access: 5 August 2019). FINNv1.5 data can be found at http://bai.acom.ucar.edu/Data/fire/ (last access: 5 August 2019).

**Technical issues:**

1. Abstract: It may be difficult for the readers who are not familiar with the Chinese geography to follow the descriptions using the province names.

**Response:** To further interpret the basic geographical profile of the focal region (i.e., CEC), we supplement two sentences to briefly introduce its inclusive provinces.

**Added/rewritten part in Abstract:** This region includes nine provinces, i.e., Hubei, Anhui, Hunan, Jiangxi, Shandong, Jiangsu, Shanghai, and Fujian. The former four ones are located inland, while the others are on the eastern coasts.

**References:**

Aouizerats, B., Van Der Werf, G. R., Balasubramanian, R. and Betha, R.: Importance of transboundary transport of biomass burning emissions to regional air quality in Southeast Asia during a high fire event, Atmos Chem Phys, 15(1), 363–373, 2015.

Binkowski, F. S. and Roselle, S. J.: Models-3 Community Multiscale Air Quality (CMAQ) model aerosol component 1. Model description, J Geophys Res Atmos, 108(D6), 2003.

Chen, Z., Chen, D., Wen, W., Zhuang, Y., Kwan, M.-P., Chen, B., Zhao, B., Yang, L., Gao, B., Li, R. and others: Evaluating the "2+ 26" regional strategy for air quality improvement during two air pollution alerts in Beijing: variations in PM 2.5 concentrations, source apportionment, and the relative contribution of local emission and regional transport, Atmos Chem Phys, 19(10), 6879–6891, 2019.

Fu, J. S., Hsu, N. C., Gao, Y., Huang, K., Li, C., Lin, N.-H. and Tsay, S.-C.: Evaluating the influences of biomass burning during 2006 BASE-ASIA: a regional chemical transport modeling, Atmos Chem Phys, 12(9), 3837–3855, 2012.

Gan, C.-M., Hogrefe, C., Mathur, R., Pleim, J., Xing, J., Wong, D., Gilliam, R., Pouliot, G. and Wei, C.: Assessment of the effects of horizontal grid resolution on long-term air quality trends using coupled WRF-CMAQ simulations, Atmos Environ, 132, 207–216, 2016.

Han, B.-S., Baik, J.-J. and Kwak, K.-H.: A preliminary study of turbulent coherent structures and ozone air quality in Seoul using the WRF-CMAQ model at a 50 m grid spacing, Atmos Environ, 218, 117012, 2019.

Huang, J., Hsu, N. C., Tsay, S.-C., Holben, B. N., Welton, E. J., Smirnov, A., Jeong, M.-J., Hansell, R. A., Berkoff, T. A., Liu, Z. and others: Evaluations of cirrus contamination and screening in ground aerosol observations using collocated lidar systems, J Geophys Res Atmos, 117(D15), 2012.

Jeon, W., Choi, Y., Percell, P., Souri, A. H., Song, C.-K., Kim, S.-T. and Kim, J.: Computationally efficient air quality forecasting tool: implementation of STOPS v1. 5 model into CMAQ v5. 0.2 for a prediction of Asian dust, Geosci Model Dev, 9(10), 3671–3684, 2016.

Malm, W. C., Sisler, J. F., Huffman, D., Eldred, R. A. and Cahill, T. A.: Spatial and seasonal trends in particle concentration and optical extinction in the United States, J Geophys Res Atmos, 99(D1), 1347–1370, 1994.

Park, R. S., Song, C. H., Han, K. M., Park, M. E., Lee, S.-S., Kim, S.-B. and Shimizu, A.: A study on the aerosol optical properties over East Asia using a combination of CMAQ-simulated aerosol optical properties and remote-sensing data via a data assimilation technique, Atmos Chem Phys, 11(23), 12275–12296, 2011.

Qiao, X., Guo, H., Tang, Y., Wang, P., Deng, W., Zhao, X., Hu, J., Ying, Q. and Zhang, H.: Local and regional contributions to fine particulate matter in the 18 cities of Sichuan Basin, southwestern China, Atmos Chem Phys, 19(9), 5791–5803, 2019.

Song, C. H., Park, M. E., Lee, K. H., Ahn, H. J., Lee, Y., Kim, J. Y., Han, K. M., Kim, J., Ghim, Y. S. and Kim, Y. J.: An investigation into seasonal and regional aerosol characteristics in East Asia using model-predicted and remotely-sensed aerosol properties, Atmos Chem Phys, 8(22), 6627–6654, 2008.

Tai, E., Jimenez, M., Nopmongcol, O., Wilson, G., Mansell, G., Koo, B. and Yarwood, G.: Boundary conditions and fire emissions modeling, Prep Texas Comm Environ Qual Sept, 2008.

Wu, Y., Wang, P., Yu, S., Wang, L., Li, P., Li, Z., Mehmood, K., Liu, W., Wu, J., Lichtfouse, E. and others: Residential emissions predicted as a major source of fine particulate matter in winter over the Yangtze River Delta, China, Environ Chem Lett, 16(3), 1117–1127, 2018.

Xing, J., Ding, D., Wang, S., Zhao, B., Jang, C., Wu, W., Zhang, F., Zhu, Y. and Hao, J.: Quantification of the enhanced effectiveness of NO x control from simultaneous reductions of VOC and NH 3 for reducing air pollution in the Beijing--Tianjin--Hebei region, China, Atmos Chem Phys, 18(11), 7799–7814, 2018.

Yu, S., Li, P., Wang, L., Wu, Y., Wang, S., Liu, K., Zhu, T., Zhang, Y., Hu, M., Zeng, L. and others: Mitigation of severe urban haze pollution by a precision air pollution control approach, Sci Rep, 8(1), 8151, 2018.